



# 1  OMI surface UV irradiance in the continental United States: quality
# 2  assessment, trend analysis, and sampling issues

Huanxin Zhang[1,2], Jun Wang[1,2], Lorena Castro García[1,2], Yang Liu[3], Nickolay A. Krotkov[4]
[1]Department of Chemical and Biochemical Engineering, The University of Iowa, Iowa City, IA, USA
[2]Center for Global and Regional Environmental Research, The University of Iowa, Iowa City, IA, USA
[3]Rollins School of Public Health, Emory University, Atlanta, GA, USA
[4]NASA Goddard Space Flight Center, Greenbelt, MD, USA
*Correspondence to*: Jun Wang (jun-wang-1@uiowa.edu), Huanxin Zhang (huanxin-zhang@uiowa.edu)
**Abstract.** Surface full-sky erythemal dose rate (EDR) from Ozone Monitoring Instrument (OMI) at both satellite overpass
time and local noon time are evaluated against ground measurements at 31 sites from USDA UV-B Monitoring and Research
Program over the period of 2005–2017. We find that both OMI overpass time and local solar noon EDR are highly
correlated with the measured counterparts (R = 0.88). Although the comparison statistics are improved with longer time
window used for pairing surface and OMI measurements, OMI data overall has ~4% underestimate for overpass EDR while
~8% overestimate for the solar noon time EDR. The biases are analyzed regarding the spatial and temporal data collocation,
the effects of solar zenith angle (SZA), clouds and the assumption of constant atmospheric conditions. The difference
between OMI overpass EDR and ground observation shows some moderate dependence on SZA and the bias could be up to
−30 % with SZA greater than ~65°. In addition, the ratio of EDR between solar noon to overpass time is often (95% in
frequency) larger than 1 from OMI products; in contrast, this ratio from ground observation is shown to be normally
distributed around 1. This contrast suggests that the current OMI surface UV algorithm would not fully represent the real
atmosphere with the assumption of a constant atmospheric profile between noon and satellite overpass times. The viability of
surface UV in terms of peak UV frequency is also studied. Both OMI Noon_FS and ground peak EDR show a high
frequency of occurrence of ~ 20 mW m$^{-2}$ over the period of 2005–2017. However, another high frequency of ~ 200 mW m$^{-2}$
occurs in OMI solar noon EDR while the ground peak values show the high frequency around 220 mW m$^{-2}$, implying that
the OMI solar noon time may not always represent the peak daily UV values. Lastly, OMI full-sky solar noon EDR shows
statistically significant positive trends in parts of the northeastern U.S., the Ohio River Valley region and California.
However, the UV trends estimated from ground-based network using two sampling methods (one corresponds to the OMI
noon time and one averages all the data in a day) show significant negative trends in the Northeast and the Ohio River Valley
region, which is consistent with the increase of absorption aerosol optical depth as revealed by OMI aerosol product in these
regions. No statistically-significant trend can be found for OMI columnar O3 or cloud optical depth. The future surface UV
data estimated with better spatial and temporal resolution obtained from geostationary satellites would help resolve these
discrepancies found in the biases and estimated surface UV trends.



## 1 Introduction

The amount of surface solar UV radiation (200–400 nm) reaching the earth's surface has substantial impacts on human health and ecosystems (UNEP, 2007; WMO, 2010). For example, about 90 % of nonmelanoma skin cancers are associated with exposure to solar UV radiation in the United States (Koh et al., 1996). Bornman and Teramura (1993) and Caldwell et al. (1995) showed the negative effects of UV radiation on plant growth and tissues. Since the discovery of the significant ozone depletion in the Antarctic region (Farman et al., 1985) and mid latitudes (Fioletov et al., 2002), subsequent effects on surface UV levels have received attention. As a result, great efforts have been made to monitor surface UV radiation from both satellite and ground instruments in the past few decades (Bigelow et al., 1998; Sabburg et al., 2002; Levelt et al., 2006). Although satellite measurements provide a better spatial coverage of the surface UV radiation, they (similar to ground-based observations) are not only affected by instrument errors (Bernhard and Seckmeyer, 1999), but are also subject to uncertainties in the algorithms used to derive surface UV radiation. Therefore, evaluation of satellite-based estimates of surface UV radiation against available ground measurements in many locations around the world is needed to characterize the errors toward further refinement of the surface UV estimates.

The solar spectral irradiance (in mW m$^{-2}$ nm$^{-1}$) is usually measured by ground and satellite instruments. In addition, the surface UV irradiance, denoted as 'erythmal weighted', has been widely used to describe the sunburning or reddening effects (McKenzie et al., 2004). Erythemally weighted irradiance or erythemal dose rate (in mW m$^{-2}$) is defined as the incoming solar radiation on a horizontal surface weighted according to the erythemal action spectrum (McKinlay and Diffey, 1987) ; it can be further divided by 25 mW m$^{-2}$ to derive UV index - an indicator of the potential for skin damage (WMO, 2002). Hence, UV index is commonly used as a UV exposure measure to the general public and in epidemiological studies in the U.S. and other parts of the world (Eide and Weinstock, 2005; Lemus-Deschamps and Makin, 2012; Walls et al., 2013). In the U.S., several ground UV monitoring networks have been established responding to changes in the surface UV radiation (Bigelow et al., 1998; Sabburg et al., 2002; Scotto et al., 1988). Currently, the UVMRP initiated by the USDA remains as the only active and largest operating network providing climatological surface UV data in the United States.

The goal of this study is to use UVMRP datasets to evaluate the OMI-based estimates of the surface UV radiation in the past decade in the United States. As a successor of Total Ozone Mapping Spectrometer (TOMS) whose surface UV data (such as erythemally weighted irradiance) has been extensively evaluated in the past (Arola et al., 2005; Cede et al., 2004; Kalliskota et al., 2000; Kazantzidis et al., 2006; McKenzie et al., 2001), the OMI data has a much finer spatial and spectral resolution and thereby bears more advanced capability for characterizing the spatial distribution of the surface UV radiation. TOMS data records span from 1978 to 2005, and many past studies have shown that TOMS surface UV data overestimated the ground observational data in many sites. OMI was launched into space in July 2004 as part of the Aura satellite (Levelt et al., 2006), and it has started to collect data from August 2004 to the present. While there have been a number of studies



evaluating the OMI surface UV data with ground observations, these studies, as shown in Table 1, have mainly focused on
Europe (Antón et al., 2010; Buchard et al., 2008; Ialongo et al., 2008; Kazadzis et al., 2009; Tanskanen et al., 2007; Weihs et
al., 2008; Zempila et al., 2016), high latitudes (Bernhard et al., 2015) and the tropics (Janjai et al., 2014). These studies
evaluated OMI spectral irradiance, EDR and erythemally weighted daily dose within different time periods. Most
comparisons show positive bias up to 69 % with few show negative bias up to –10 %.

This study differs from the past studies in the following ways. Firstly, we conducted a comprehensive evaluation of the OMI
surface UV data from 2005 to 2017 covering the continental United States. The evaluation was made for erythemally
weighted irradiance at both local solar noon and satellite overpass times, and the evaluation statistics not only concern mean
bias but also the probability density function (PDF), cumulative density function (CDF) and variability of the UV data.
Secondly, a trend analysis of the surface UV irradiance from both ground observation and OMI was performed, with a
special focus on the effects of the temporal sampling. The analysis addresses if the once-per-day sampling from the polar-
orbiting satellite would have any inherent limitation for the trend analysis of surface UV data. Finally, the error
characteristics in the OMI surface UV data were examined to understand the underlying sources (such as from treatment of
clouds and assumption of constant atmospheric conditions between the local solar noon and satellite overpass time). The
investigation yields recommendations for future refinement of the OMI surface UV algorithm.

The paper is organized as follows: Sect. 2 describes the satellite and ground observational data; the methodology is discussed
in Sect. 3; Sect. 4 presents the results and Sect. 5 summarizes the findings.

**2 Data**
**2.1 OMI data**
OMI aboard the NASA Aura spacecraft is a nadir-viewing spectrometer (Levelt et al., 2006) that measures solar reflected
and backscattered radiances in the range of 270 nm to 500 nm with a spectral resolution of about 0.5 nm. The 2600 km wide
viewing swath and the sun-synchronous orbit of Aura provides a daily global coverage, with an equatorial crossing time at ~
13:45 local time. The spatial resolution varies from 13 x 24 $km^2$ (along x cross) at nadir to 50 x 50 $km^2$ near the edge. OMI
retrieves total column ozone, total column amount of trace gases $SO_2$, $NO_2$, HOCO, aerosol characteristic and surface UV
(Levelt et al., 2006).

The OMI surface UV algorithm has its heritage from the TOMS UV algorithm developed at NASA Goddard Space Flight
Center (GSFC) (Eck et al., 1995; Herman et al., 1999; Krotkov et al., 1998; Krotkov et al., 2001; Tanskanen et al., 2006;
Krotkov et al., 2002). In the first part of the algorithm, the surface-level UV irradiance at each OMI pixel under clear-sky



conditions is estimated from a look-up table that is computed from a radiative transfer model for different values of total
column ozone, surface albedo, and SZA. The look-up table was called twice, once to calculate the surface UV irradiance at
the satellite overpass time and once at the local solar noon. The only difference between these two look-up tables are the
SZAs with one representing the SZAs at the overpass time and the other representing the solar noon, while the total column
ozone and cloud optical thickness (COT) are assumed to stay constant. The second step is to correct the clear-sky surface
UV irradiance for a given OMI pixel due to the effects of cloud and non-absorbing aerosols. The cloud correction factor is
derived from the ratio of measured backscatter irradiances and solar irradiances at 360 nm along with OMI total column
ozone amount, surface monthly minimum Lambertian Effective Reflectivity (LER), and surface pressure. The effects of
absorbing aerosols are also adjusted in the current surface UV algorithm based on a monthly aerosol climatology as
described in Arola et al. (2009).

The second step of the cloud correction mentioned above follows radiative transfer calculations that assume a homogeneous,
plane parallel water-cloud model with Rayleigh scattering and ozone absorption in the atmosphere (Krotkov et al., 2001).
The COT is assumed to be spectrally independent and the cloud phase function follows the C1-cloud model (Deirmendjian,
1969). This cloud model is also used to calculate the angular distribution of 360 nm radiance at the top of the atmosphere,
which is used to derive an effective COT. The effective COT is the same as the actual COT for a homogeneous cloud plane-
parallel model. The effective COT is saved to a look-up table to use for cloud correction.

OMI surface UV data products (or OMUVB in shorthand) include: (a) spectral irradiance (mW m$^{-2}$ nm$^{-1}$) at 305, 310, 324
and 380 nm at both the local solar noon and OMI overpass time, (b) erythemal dose rate (EDR, mW m$^{-2}$) at both the local
solar noon and OMI overpass time and (c) erythemally weighted daily dose (EDD, J m$^{-2}$). The spectral irradiances assume
triangular slit function with full width half maximum of 0.55nm. The EDD is computed by applying the trapezoidal
integration method to the hourly EDR with the assumption that the total column ozone and COT remain the same throughout
the day. In addition, the OMUVB products include information on data quality related to row anomaly, SZA and COT which
are used in the present study. We also use the aerosol products from the OMAERUV algorithm (Torres et al., 2007). The
OMI OMAERUV algorithm uses two wavelengths in the UV region (354 and 388 nm) to derive aerosol extinction and
absorption optical depth. The aerosol products (OMAERUV) retrieve aerosol optical depth (AOD), aerosol absorption
optical depth (AAOD) and single scattering albedo at 354 nm, 388 nm and 500 nm.

In the current study, both OMI level 2 (v003) and level 3 (v003) products are used. The level 2 provides swath level data
products while level 3 products are gridded daily products on a 1º x 1º horizontal grid. Two variables from OMUVB level 2
products (Table 2) are used: 1) full-sky solar noon erythemal dose rate denoted as Noon_FS EDR; 2) full-sky overpass time
erythemal dose rate denoted as OP_FS EDR. In addition, full-sky solar noon EDR from the OMUVBd (d denotes daily)
level 3 products and AOD and AAOD from OMAERUVd level 3 products are used. These level 3 datasets are mainly used



for conducting trend analysis in Sect. 4.4 unless noted otherwise while the rest of the data analysis use the level 2 datasets.
All the datasets are from January 2005 to December 2017 and row anomaly is checked during data analysis for level 2
datasets.

## 2.2 Ground observation data

Currently, the UVMRP operates 36 climatological sites for long-term monitoring of surface UV radiation around different
ecosystem regions (https://uvb.nrel.colostate.edu/UVB/uvb-network.jsf). Of the 36 climatological sites, five are located in
New Zealand, South Korea, Hawaii, Alaska and Canada, while 31 sites are in the continental U.S., with the majority of them
located in agricultural or rural areas and a few in urban areas. Among these 31 sites, one site started operation after 2014 and
one after 2006, and all other sites started earlier than 2006. In the current study, we use the one site in Canada and 30 of the
31 sites in the continental U.S. and we exclude one site where operation started after 2014 (Fig. 1). All sites measure global
irradiance in the UVB spectral range (280–320 nm), using a UVB1-pyranometer manufactured by Yankee Environmental
Systems (YES). The YES UVB-1 instrument takes measurement every 15 seconds which are aggregated into 3-min
averages. These output data are calibrated following Lantz et al. (1999) and weighted according to McKinlay and Diffey
(1987) to generate the erythemally weighted irradiance (300–400 nm). The calibration and characterization of each YES
pyranometer were performed annually. The pyranometers differ from the collocated standard triad within ~ ±2.8 % for SZA
< 80° and the absolute calibration uncertainty errors could reach ~ ±10 % in some cases when SZA is > 80° (Bigelow et al.,
1998; Lantz et al., 1999). In spite of this, McKenzie et al. (2006) has shown that the relative uncertainties could be more
important when evaluating the geographical differences in erythemal weighted irradiance at mid-latitude sites maintained by
USDA. In this work, we use the 3-min averaged erythemally weighted irradiance at 31 sites in the continental U.S. and
information for each site is described in Table 3. Except for site TX41, for which data were available since August 2006, we
use data from January 2005 to December 2017 for the rest of the sites.

## 3 Methods

### 3.1 Spatial collocation and temporal averaging of data

Since OMI data represent an average over a ground pixel (~13 x 24 km$^2$ for nadir viewing and ~50 x 50 km$^2$ for off-nadir
viewing) and ground measurements are point measurements that cover a small area, previous work in Table 1 has
investigated the effects of the selection of a collocation distance between the center of an OMI ground pixel and the ground
observational site or the averaging time period around OMI overpass time/local solar noon on the evaluation results. For
example, Weihs et al. (2008) found the variability, defined as the absolute sum of the difference between the average mean
bias between OMI and ground measured UV index at any station and the average mean bias from all stations divided by the
total number of measurements, increases with increasing collocation distance but decreases with increasing averaging time



period. Zempila et al. (2016) compared OMI spectral irradiances at 305, 310, 324 and 380 nm with ground observations
considering different spatial collocation and temporal averaging windows. It was shown that the choice of collocation
distance (10 km, 25 km or 50 km) plays a negligible role in the comparison in terms of the correlation coefficient and mean
bias. However, the selection of longer averaging time period (from ± 1 minute to ± 30 minutes) results in a significant
improvement under full-sky conditions for both OMI overpass and solar noon time comparison. (Chubarova et al., 2002)
evaluated the difference between TOMS overpass surface UV and ground data taken over different time windows around
TOMS overpass time. The results showed that the calculated correlation coefficient of these two datasets nonlinearly
increases with the increasing averaging windows (from ± 1 minute to ± 60 minutes) and stays nearly constant from ± 60
minutes to ± 90 minutes.

In this work, we will examine the separate effects of spatial collocation and temporal averaging on evaluation results. Firstly,
for each ground site, its observation is paired with the OMI data at pixel-level if the center of that pixel is within the distance
(D) of 50 km from that ground site. Then the ground observational data at each site is taken within (ΔT of) ± 5 minutes
around the OMI overpass time or the local solar noon time at that pixel. Correspondingly, the temporal mean of ground
observation within ΔT is compared to the spatial mean of OMI data within D. Further evaluation is conducted by changing
different D values to 10 km and 25 km and/or ΔT values of ± 10, ± 30 and ± 60 minutes around OMI overpass time and local
solar noon time. Consequently, a total of 12 sets of paired data are generated for the evaluation, as a result of a different
combination of three D values and four ΔT values used for spatially and temporally collocating OMI and ground data. For a
given ΔT, there are ~ 100,000, ~ 67,000, ~ 17,000 data pairs for D values of 50 km, 25 km and 10 km respectively.
**3.2 Validation statistics**
First, we present several commonly used validation statistics (Table 2): Mean Bias (MB) calculated in Eq. (1), normalized
mean bias (NMB, %) in Eq. (2), the root-mean-square error (RMSE) in Eq. (3) and correlation coefficient (R). We also show
the overall evaluation of OMI surface UV data against ground observation in the form of a Taylor Diagram (Taylor, 2001)
(see Fig. 3(a)). Taylor Diagram provides a statistic summary of OMI data evaluated against ground observation in terms of
correlation coefficient R (the cosine of polar angles), the ratio of standard deviations between OMI and ground observational
data (the normalized standard deviation (NSD)) shown in x and y axis respectively, and the normalized room-mean-square
difference (RMSD), shown as the radius from the expected point, which is located at the point where R and NSD are unity.
The following equations are represented:
$MB = \frac{1}{N}\sum_{i=1}^{N}(EDR_{(OMI,i)} - EDR_{(Ground,i)}),$ (1)
$NMB = \frac{\sum_{i=1}^{N}(EDR_{(OMI,i)} - EDR_{(Ground,i)})}{\sum_{i=1}^{N}EDR_{(Ground,i)}},$ (2)
$RMSE = \sqrt{\frac{1}{N}\sum_{i=1}^{N}(EDR_{(OMI,i)} - EDR_{(Ground,i)})^2},$ (3)



Where i is the i-th paired (OMI-Ground) data point, N is the total number of paired data points and $EDR_{(OMI,i)}$ and
$EDR_{(Ground,i)}$ are the ith EDR from OMI and ground observation, respectively.

To determine whether the calculated MB or NMB are statistically significant, a t-test for differences of mean under serial
dependence is applied (Wilks, 2011). This two-sample t-test assumes a first-order autoregression in the data. The computed
two-tailed p-value of less than 0.025 indicates that the difference between the means for the paired data (OMI and ground
EDR) would be statistically significant at the 95% confidence level. In addition, we calculate the PDF and CDF of the OMI
and ground observation. A Kolmogorov-Smirnov (K-S) test (Wilks, 2011) is performed to compare the CDFs of the OMI
and ground datasets. The K-S test is represented by the following formula:
$D = max\ |CDF_{OMI} - CDF_{Ground}|,$         (4)
If D is greater than the critical value, $0.84\sqrt{1/n}$ (n is the total number of data points), then the null hypothesis that the two
datasets were drawn from the same distribution will be rejected at the 99 % confidence level.

**3.3 Trend analysis**

Following the work of Weatherhead et al. (1997) and Weatherhead et al. (1998), the trend of surface UV irradiance from
OMI and ground observation can be estimated using the following linear model:
$Y_t = C + S_t + \omega X_t + N_t$      t = 1… T,         (5)
Where T is the total number of months considered and t is the month index, starting from January 2005 to December 2017.
$Y_t$ is the monthly mean surface UV irradiance either from OMI or the ground observation in the U.S. and C is a constant. $X_t$
= t/12, represents the linear trend function and $\omega$ is the magnitude of the trend per year. $S_t$ is a seasonal component,
represented in the following form:
$S_t = \sum_{j=1}^{4} [\beta_{1,j} \sin(2\pi jt/12) + \beta_{2,j} \cos(2\pi jt/12)],$         (6)
$N_t$ is the noise not represented by the linear model and is often assumed to be a first-order autoregressive model, which can
be expressed as:
$N_t = \phi N_{t-1} + \varepsilon_t,$         (7)
Where $N_{t-1}$ is the noise from month (t-1), $\phi$ is the autocorrelation between $N_t$ and $N_{t-1}$, $\varepsilon_t$ is the white noise which should
be approximately independent, normally distributed with zero mean and common variance $\sigma_\varepsilon^2$.
As described in Weatherhead et al. (1998), General Least Squares (GLS) regression was applied to equation (5) to derive the
approximation of ω and its standard deviation $\sigma_\omega$ as
$\sigma_\omega = \frac{\sigma_N}{n^{3/2}} \sqrt{\frac{1+\phi}{1-\phi}},$         (8)
Where n = T/12, is the number of years of the data used in the analysis and $\sigma_N$ is the standard deviation of $N_t$. We will
consider the trend significant at the 95 % confidence level if $\omega/\sigma_\omega > 2$. Such linear models have been widely used to study



## 4 Results

### 4.1 Spatial and temporal inter-comparison

the various environmental monthly time series data in the previous studies (Boys et al., 2014; Zhang and Reid, 2010;
Weatherhead et al., 2000).
**4 Results**
**4.1 Spatial and temporal inter-comparison**
Figure 1 shows the map of OMI level 3 EDR at solar noon time under full-sky conditions averaged from 2005–2017,
overlaid with 31 ground observational sites of EDR averaged from the same local noon time. First, we find that OMI data
shows a meridional gradient with the dose rate increasing from ~ 80 mW m$^{-2}$ in the northern U.S. to ~ 203 mW m$^{-2}$ in the
southern U.S. At higher elevation regions such as in Colorado, OMI-derived EDR are larger than other areas of the same
latitude zone. In comparison, the ground sites range from ~ 71 mW m$^{-2}$ in the northern U.S. to a maximum of ~ 200 mWm$^{-2}$
for site NM01 in the southern U.S., generally capturing the OMI meridional gradient well. At most sites, OMI data
overestimates the ground observation by more than 5 %, with sites in Steamboat Spring, Colorado (CO11), Burlington,
Vermont (VT01) and Homestead, Florida (FL01) showing the highest bias of more than 15 %.
Scatter plots of OMI OP_FS and Noon_FS EDR with all 31 ground observational sites are shown in Fig. 2(a) and (b). In
both cases, a linear relationship is found with correlation coefficient (R) of 0.88. This statistically significant correlation
(with P < 0.01) can also be found at most individual sites, as shown in the Taylor Diagrams (Fig. 3(a) and (b)). The high
correlation found here in the U.S. is consistent with previous work that evaluated OMI EDR in Europe (Buchard et al., 2008;
Ialongo et al., 2008). However, lowest R of 0.66 and 0.65 at Florida (FL01) are found respectively for OMI OP_FS and
Noon_FS EDR (shown in Fig. 2(c) and (d)). Even though both OMI OP_FS and Noon_FS EDR data show good correlation,
their differences show different signs and magnitudes. Overall, we find that the MB for OMI OP_FS EDR comparison is −
4.1 mW m$^{-2}$ while the MB for OMI Noon_FS EDR comparison is 10.1 mW m$^{-2}$. The respect RMSE values are 39.8 and 42.2
mW m$^{-2}$. Figure 3 (a) and (b) show the evaluation of OMI OP_FS and Noon_FS EDR with D = 50 km and ΔT = ± 5 minutes
for 31 ground observational sites in the form of a Taylor Diagram and Fig. 4(a) and (b) are the corresponding zoomed-in
plots. As can be seen, with the case of OMI OP_FS EDR evaluation, 26 sites have negative NMB ranging from −14 % to −
1.5 % with 16 sites being statistically significant at 95 % confidence level. Steamboat Springs, Colorado (CO11),
Homestead, Florida (FL01) and Burlington, Vermont (VT01) show statistically significant (95 % confidence level) positive
bias. The site in Holtville, California (CA21) shows no significant difference between OMI OP_FS EDR and ground
observation. For OMI Noon_FS EDR, the majority of the sites have positive NMB (3–31 %) with site Steamboat Springs,
Colorado (CO11) having the largest NMB of 31 %. The NMB found in most of the sites show significant difference at the 95
% confidence level except for sites in Holtville, California (CA21), Georgia (GA01) and Utah (UT01). With both datasets,
the site at CO01 show high positive bias because of its high altitude (~ 3 km). The current OMI surface UV algorithm does
not use any cloud correction for altitudes higher than 2.5 km, which leads to a clear-sky condition for higher altitudes.




The NSD of evaluating OMI OP_FS EDR for the majority of the sites varies from 0.75 to 1 (Fig. 4(a)), indicating that the
OMI OP_FS EDR underestimates the amplitude of surface UV irradiance cycle found in the ground observation. In contrast,
we find from Fig. 4(b) that about half of the ground sites have NSD values ranging from 1 to 1.1 while the rest of the sites
have NSD values less than 1 for OMI Noon_FS EDR evaluation. In both cases, the ground site at Raleigh, North Carolina
(NC01) has the lowest NSD of ~ 0.75. Additionally, sites in the southeastern U.S. (e.g., FL01, LA01, GA01, NC01) along
with the site in Houston, Texas (TX41) all have relatively larger RMSD (greater than 0.5) for both OMI OP_FS and
Noon_FS EDR evaluation according to Fig. 4(a) and (b), respectively. In comparison, sites in the northern higher latitude
seem to show smaller RMSD (e.g., WA01, NE01, NY01, ND01). Overall, the site in Davis, California (CA01) show the best
performance in terms of R, NSD and RMSD for both OMI OP_FS and Noon_FS EDR evaluation. These regional differences
reflect the effects of the spatial variability of U.S. climate and air pollution on surface UV estimates. The southeastern U.S.
is subject to heavy pollution and this region is largely affected by clouds. This could pose a greater challenge for the OMI
surface UV algorithm. These discrepancies can be related to several factors such as the method of collocating OMI data with
ground observation spatially and temporally, clouds in the atmosphere, and the assumption of constant atmospheric
conditions between OMI overpass time and local solar noon time, which are discussed in the following sections.

To further show how well OMI surface EDR represents the ground observational EDR, the PDFs of both OMI and ground
EDR are shown (Fig. 5). First, we find the distribution of surface EDR at solar noon time from both OMI and ground
observational data show two peaks, one around 20 mW m$^{-2}$ and the other one around 200 mW m$^{-2}$. Similar distribution with
two peaks are also found for OMI and ground EDR at overpass time which are not shown here. These two peaks are largely
due to the SZA effects, with the former one related to larger SZAs and the latter one with smaller SZAs. The work of Wang
and Christopher (2006) also indicated that the change in SZA causes the solar downward shortwave irradiance to show two
peaks one at ~08:00 LT and another one at ~16:00 LT. Figure 6 show the calculated CDFs for OMI and ground OP_FS and
Noon_FS EDR as well as the maximum difference between EDRs at the corresponding time. The critical values for both
comparisons are 0.087 to verify that the two CDFs show a good fit at the 99 % confidence level. From Fig. 6, we can see that
both of the maximum differences are smaller than the critical values at the 99 % confidence level. Therefore, the null
hypothesis (OMI surface EDR and ground observed EDR were drawn from the same distribution) will not be rejected. This
good fit between OMI and ground EDR distribution for both solar noon time and overpass time again confirms the good
correlation found between these two datasets.

In order to better understand the variability of surface UV, the peak UV frequency inferred from ground observation is
investigated along with OMI Noon_FS EDR frequency. As seen in Fig. 7, both OMI Noon_FS and ground peak EDR show a
high frequency at the lower end of surface EDR (< 100 mW m$^{-2}$), which also reflects the smaller peak found in Fig. 5.
Moreover, this high frequency of occurrence persisted from 2005 to 2017 for both datasets. In addition, OMI Noon_FS EDR



shows another high frequency of surface EDR around 200 mW m$^{-2}$ corresponding to the other peak in Fig. 5. However, the
ground peak does not capture this high frequency occurrence of ~ 200 mW m$^{-2}$, instead, the ground peak values find a high
frequency around ~ 220 mW m$^{-2}$ (shown in the red box in Fig. 7). This indicates that the OMI solar noon time EDR may not
always represent the high peak value on a daily basis due to the varying atmospheric conditions. The high frequency
occurrence of ~ 220 mW m$^{-2}$ prevailed until 2015, at the same time, we find the frequency of higher surface EDR from
ground peak of ~ 300 mW m$^{-2}$ starts to increase around 2014 (shown in the red box in Fig. 7). This increase in the
occurrence of peak UV intensity could have potential implications for human exposure and subsequent health effects, which
is beyond the scope of this study.

**4.2 Impacts of spatial collocation and temporal averaging**

Table 4 and Table 5 summarize the regression statistics and other validation statistics of evaluating OMI OP_FS and
Noon_FS EDR with different spatial collocation distances (D) and temporal averaging windows (ΔT), respectively. We find
that the spatial collocation distances do not affect the overall comparison results significantly. Even though the stricter
collocation distance within 10 km radius (or D = 10 km) results in 41 % decrease in MB for OMI OP_FS EDR evaluation,
the collocated data sample size is reduced to only about 17 % of the original datasets. In contrast, the length of temporal
averaging window seems to play a more important role in the overall comparison results. Figure 4 (a) to Figure 4(c) show
that most of the dots representing the OMI OP_FS EDR evaluation on the Taylor Diagram are moving closer to the expected
point as ΔT increases from ±5 minutes to ±60 minutes. The same progression is also found for OMI Noon_FS EDR
evaluation which is not shown here. Specifically, the increasing temporal windows cause the NSD values to increase. On the
other hand, R increases and RMSD decreases as temporal average window ΔT increases from ±5 minutes to ±60 minutes in
both cases, which can be also found in Fig. 4(d). Moreover, the RMSE values decrease by about 16.8 % and 11.1 % as ΔT
increase from ±5 minutes to ±60 minutes for OP_FS and Noon_FS EDR comparison, respectively. The improvement with a
longer temporal averaging window for overpass time under full-sky is also found by Zempila et al. (2016). Additionally,
changes in the sign of NMB from negative to positive are found at some of the sites for OMI OP_FS evaluation when ΔT
increases from ±5 minutes to ±60 minutes. The positive NMB is significant for sites CA21, TX41, MS01, ME01, MT01 and
VT01. This could suggest that atmospheric conditions do not stay the same over this longer temporal averaging window.

**4.3 Impacts of the assumption of constant atmospheric conditions**

As described in Sect. 2.1, the current surface UV algorithm assumes the same atmospheric conditions at OMI overpass time
and the local solar noon time regarding cloudiness, total column ozone and atmospheric aerosol loadings but with different
SZAs. However, this assumption may not hold all the time for the real atmosphere. We take the ratio between Noon_FS and
OP_FS EDR (Noon_FS/OP_FS) from both OMI and ground data as an indicator of the variation of atmospheric conditions
between these two times. Figure 8 shows the frequency and PDF of this ratio from both OMI and ground data obtained with



D = 50 km and ΔT = ±5 minutes. The ground ratio is approximately equally distributed around the center of 1 while about 95
% of the OMI data falls into the area with the ratio greater than 1. This indicates that the current OMI surface UV algorithm
would not fully represent the real atmosphere with the assumption of constant atmospheric conditions being made and could
thus induce errors in estimating surface UV irradiances. The scatter plot of the ground ratio and OMI ratio further confirms
the inconsistency between the OMI data and the observational data (Fig. 9) with no significant correlation being found.

We further investigate the possible seasonal effects on this ratio. As can be seen in Fig. 10, the mean and median ratio
(Noon_FS/OP_FS) from OMI are greater than those from the ground observational data throughout the year, which again
indicates the potential overestimation of OMI Noon_FS EDR using constant atmospheric conditions. Furthermore, the
discrepancy between these two ratios stays consistent in the spring and summer time. The smaller SZA in the summer time
would have relatively small effects and the difference in these ratios could be largely affected by the varying atmospheric
conditions between local solar noon time and OMI overpass time. However, this discrepancy becomes larger in the fall and
winter time, which could be the result of the elevated SZA towards winter time in North America to some extent. The larger
SZA (> 70°) in the colder times could increase the radiation path in the atmosphere which would thereby amplify the
atmospheric interaction with the solar radiation. Besides, other seasonal variables such as the climatological albedo used in
the current OMI surface UV algorithm could potentially play a role in the deviation between OMI and ground data. In
addition, the ratio from both OMI and ground observational data show larger variation in the fall and winter season than its
respective summer season, implying the impacts of the SZA seasonal variation on both OMI and observational data.

The SZA seasonal variation could subsequently affect the difference between OMI and ground data, which will be analyzed
in this section. Several previous studies have investigated the effects of SZA on the difference between OMI and ground
observational irradiance. Buchard et al. (2008) found that OMI spectral UV irradiance on clear-sky days showed a larger
discrepancy at SZA greater than 65°. Kazadzis et al. (2009) found no systematic dependence of the difference between OMI
and ground observational spectral UV irradiance on SZA. By sorting data based on cloud and aerosol conditions, Antón et al.
(2010) showed that the relative difference between OMI and ground irradiance decreases modestly with SZA for all-sky
conditions except for days with high aerosol loadings. Zempila et al. (2016) suggested a small dependence of the ratio
(OMI/ground UV irradiance) on SZA under both clear-sky and all-sky conditions. For the all-sky condition, the ratio
increases steadily with increasing SZA up to 50° and becomes larger than one after 50°. From the simple regression derived
using bin averaged data (Fig. 11), we find that the OMI OP_FS EDR bias has a stronger dependence on the overpass SZA
than Noon_FS EDR. At smaller SZAs, the median of OMI OP_FS EDR bias show smaller dependence, however, the median
increases greatly (up to −30%) when SZA is greater than ~ 65 °.

Clouds also play an important role in the difference between OMI and ground observational UV irradiance. Buchard et al.
(2008) found that the relative difference between OMI and ground EDR was associated with COT at 360 nm retrieved from



OMI and the difference is more appreciable for large COT. Tanskanen et al. (2007) showed that the distribution of the OMI
and ground EDD ratio widens with increasing COT. Antón et al. (2010) used OMI retrieved LER at 360 nm as a proxy for
cloudiness and showed that the relative difference of OMI and ground EDR increased largely at higher LER values. Here,
we find that the relative bias for OMI OP_FS EDR is more obvious at larger COT values as well (Fig. 12). In addition, the
noise of the bias gets larger at higher COT values. This is due to the fact that OMI surface UV algorithm uses the average of
a pixel to represent the cloudiness in that specific pixel. In reality, the spatial distribution of cloudiness in that pixel could
vary a lot which could result in the large difference in surface UV irradiance between the OMI pixel and the ground
observational site.
**4.4 Trend analysis**
EDR is the weighted solar irradiance from 300–400 nm which covers the UVB range principally controlled by the
atmospheric ozone column. In addition, both UVA and UVB could be affected by the cloud cover and aerosol loadings in the
atmosphere. Thus, the identified trend of surface EDR could be a result of the combined effects of the aforementioned
different factors and it would be challenging to attribute the trend to any individual factor quantitatively. Therefore, we focus
on providing a descriptive summary of surface EDR trends derived from both OMI and ground observation.

We first analyze the surface EDR trend using OMI level 3 data. We find that OMI full-sky solar noon EDR data show a
positive trend in most of the places; but the only significant trend (95 % confidence level) was found in parts of the
northeastern U.S., in parts of the Ohio River valley region and in a small part of California (Fig. 13(b)). A similar
distribution of trend is found in OMI level 3 full-sky spectral irradiance at 310 nm (Fig. 14 (a)). We also analyzed the trend
of OMI level 3 clear-sky EDR and total column ozone amount (not shown here) and found no significant trend in either
dataset. This could suggest that the contribution of ozone column to the estimated trend of OMI full-sky EDR is minimal.
Instead, the estimated trend could be induced by other factors such as changes in the local cloudiness and absorbing aerosols.
No significant trends of OMI AOD and COT are found over U.S. in this work. Zhang et al. (2017) found significant positive
trends over the western U.S. using OMI AOD for 2005–2015 and Hammer et al. (2018) found small positive trends over the
western and central U.S. with OMI AOD (388 nm) from the OMI OMAERUV algorithm for 2005–2015. However, we find
significant positive trends of OMI AAOD at 388 nm in part of the central and eastern U.S. and western U.S. close to the
coast (Fig. 14(b)). Zhang et al. (2017) found a significant increase in OMI AAOD in the southern and central U.S. and
proposed that this increase is largely caused by dust AAOD. The OMI surface UV algorithm uses a monthly mean
climatological aerosol data (Kinne, 2009), and it may not be well updated to represent the role of absorbing aerosols in
attenuating the surface UV radiation considering the diurnal and day to day variations, which may result in the contrasting
trends of OMI AAOD and surface EDR in the northeastern U.S. and the Ohio River Valley region found here.




In contrast, ground observation shows different trend patterns using two different sampling methods. For both methods, only
months with more than 10 days of data are used for trend analysis and considered missing values otherwise. The first method
is to average the ground observational data with D = 50 km and ΔT = ±5 minutes around local solar noon time, denoted as
once-per-day sampling. Eighteen of 31 sites are found to have significant trends at the 95 % confidence level (Fig. 13 (b)).
Seven sites have positive trends while the rest of the 11 sites show negative trends. The second method averages all the data
in a day at each site, hereby referred to as all-per-day sampling. We find that this method results in 15 sites with significant
trends at the 95% confidence level (Fig 13(c)). Only 4 of the 15 sites have positive trends with the rest of the sites showing
negative trends.

Both methods (e.g., once-per-day and all-per-day) find significant negative trends for sites in the Northeast and the Ohio
River Valley region with all-per-day method showing smaller trends. Using the site IN01 as an example, Figure 15 illustrates
the difference between these two sampling methods. Both methods could capture the seasonal variation of the surface EDR,
however, the magnitude of all-per-day sampling EDR is about 3 times smaller than that of the once-per-day sampling, which
is anticipated because the all-per-day average is smaller than one-per-day measurement around noon time. By averaging all
the daytime data, the all-per-day sampling method smooths out the atmospheric conditions throughout the day. In contrast,
the estimated trend of OMI Noon_FS EDR at this site is not significant, and this contrast suggests the importance to account
for the variation of atmospheric conditions throughout the daytime. The estimated positive trend from OMI AAOD at this
region could be the cause of the negative trend derived from the observed EDR, further suggesting the need to consider the
change of AAOD in estimating surface UV radiation.
**5 Conclusion and discussion**
In this study, we evaluated the OMI surface erythemal irradiance at overpass time and solar noon time for the period of
2005–2017 with 31 UVMRP ground sites in the continental United States. The OMI surface Noon_FS EDR shows a
meridional gradient with the EDR increasing from ~ 80 mW m$^{-2}$ in the northern U.S. to ~ 203 mW m$^{-2}$ in the southern U.S.
The ground observational data could capture this gradient well with EDR increasing from ~ 71 mW m$^{-2}$ in the northern U.S.
to maximum of ~ 200 mW m$^{-2}$ in the southern sites.

The comparison for both OMI OP_FS and Noon_FS EDR show good correlation with the counterparts from ground-based
measurements, with R = 0.88 when the data is matched with D = 50 km and ΔT = ±5 minutes. However, the bias differs in
signs and magnitudes. Overall, the OMI OP_FS EDR underestimates the ground observational data by –4.1 mW m$^{-2}$ while
OMI Noon_FS EDR overestimates by 10.1 mW m$^{-2}$. The RMSEs are 39.8 and 42.2 mW m$^{-2}$ respectively. The biases also
show large spatial variability. For OMI OP_FS EDR, the NMB ranges from –14 % to –1.5 % for most sites while several
sites (FL01, VT01 and CO11) show positive biases. In comparison, most sites for OMI Noon_FS EDR evaluation show



positive NMB ranging from 3 % to 31 %. Furthermore, for both OMI OP_FS and Noon_FS EDR comparison, R increases as
the temporal averaging windows ΔT increases from ±5, ±10, ±30, to ±60 minutes. When the temporal average window
reaches ±60 minutes, the OMI OP_FS EDR bias changes from negative to positive for some sites. This suggests that the
atmospheric condition does not stay consistent even within an hour, underscoring the importance of geostationary satellite
measurements. The relatively large bias and RMSE in magnitude for OMI Noon_FS EDR suggests the importance to
account for the variation of atmospheric conditions between solar noon and satellite overpass time, which cannot be resolved
by polar-orbiting satellite measurements but future geostationary satellites such as TEMPO, Sentinel-4 and GEMS should be
able to resolve this issue.

We also extended the evaluation of OMI and ground EDR by comparing the PDFs and CDFs as well as considering the peak
UV variability. First, both OMI and ground EDR distributions show two peaks, one around 20 and another around 200 mW
m$^{-2}$, mainly related to larger and smaller SZAs, respectively. The K-S test shows that the OMI and ground EDR are from the
same sample distribution at the 99 % confidence level. Both OMI Noon_FS and ground peak EDR show the high frequency
occurrence of the smaller peak (~ 20 mW m$^{-2}$) over the period of 2005–2017. However, the other high frequency occurrence
of OMI Noon_FS EDR (~ 200 mW m$^{-2}$) is not consistent with the high frequency found in ground peak values (~ 220 mW
m$^{-2}$), which again reveals that the OMI solar noon time may not always capture the peak UV values in a day, thus
highlighting the necessity for finer temporal resolution data.

Ground-based continuous measurements were used to show the effects of atmospheric variation on surface EDR. The ratio
of OMI Noon_FS / OP_FS EDR is greater than 1 for 95 % of the data points, while the ratio derived from the ground-based
data has a Gaussian distribution centered around 1. This means that the assumption of a consistent cloudiness, column ozone
amount and aerosol loadings between these two times would lead to large positive bias in the estimates of surface UV at
solar noon time, which is revealed in this study. Furthermore, we find that the OMI OP_FS EDR bias show some negative
dependence on the SZAs. Overall, the bias is smaller at smaller SZAs but increases greatly up to –30% when the SZA is
greater than ~ 65º. Additionally, the OMI OP_FS EDR bias shows slight dependence on COT. The error distribution of the
bias gets much wider at larger COT values. This error statistics suggests the importance of multiple scattering by aerosols
and clouds in the radiative transfer model, which is overlooked in the radiative transfer calculation for the current OMI's
look-up table approach to estimate surface UV.

Lastly, we investigated the surface UV trend from both OMI and ground observational data. Significant positive trends were
found in parts of the northeastern U.S., in the Ohio River Valley region and in a small part of California from OMI full-sky
data during solar noon time. In contrast, the trend from ground data depends on sampling method. The once-per-day
sampling at noon time shows larger spatial variability in the magnitude and signs of the trend while the all-per-day sampling
shows less variation in the magnitude. The all-per-day sampling method would smooth the variation in the surface UV data



that may result in a more uniform trend compared with the once-per-day sampling. The difference in the estimated trends
from these two methods is greater for sites in the western and central U.S. Analysis using ground-based observation with two
methods and OMI data reveal contrasting trend in the Northeast and in the Ohio River valley, implying the climatological
AAOD may not well account for the day to day and diurnal variations. While no discernable column ozone and COT trend
from OMI are found, decreasing trends of surface UV, as revealed by both methods using ground-based data, seem to be
consistent with the increasing trend of OMI AAOD, further suggesting the need to consider AAOD variability in estimates
of surface UV.
**Acknowledgements**
The research was funded by NASA's Aura satellite program (managed by Dr. Kenneth W. Jucks), Applied Sciences program
(managed by John A. Haynes), and Atmospheric Composition and Analysis Program (ACMAP managed by Dr. Richard
Eckman). The authors thank the OMI team for providing the surface UV and aerosol products, which can be downloaded
from the NASA Goddard Earth Sciences (GES) Data and Information Services Center (DISC) (https://disc.gsfc.nasa.gov).
We    also    thank    the    UVMRP    for    the    ground    observational    UV    data,    which    is    available    at
https://uvb.nrel.colostate.edu/UVB/uvb-dataAccess.jsf.

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

| Study | Location | OMI data[a] | Ground instrument | Time periods | Bias[b] |
|---|---|---|---|---|---|
| (Kazadzis et al. 2009) | Thessaloniki, Greece | Spectral (op) | Brewer MK III | Sep 2004–Dec 2007 | 30 % (305 nm), 17 % (324 nm), 13 % (380 nm)[c] |
| (Antón et al. 2010) | El Arenosillo, Spain | Spectral (op) | Brewer MK III | Oct 2004–Dec 2008 | 14.2 % (305 nm), 10.6 % (310 nm), 8.7 % (324 nm)[d] |
| | | EDR (op) | | | 12.3 % |
| (Zempila et al. 2016) | Thessaloniki, Greece | Spectral (op) | NILU-UV multi-filter radiometer | Jan 2005–Dec 2014 | 31 % (305 nm), 29.5 % (310 nm), 6.1 % (324 nm), 14.0 % (380 nm)[e] |
| | | Spectral (noon) | | | 33.6 % (305 nm), 28.6 % (310 nm), 5.6 % (324 nm), 13.2 % (380 nm) |
| (Buchard et al. 2008) | Villeneuve d' Ascq, France | EDR (op) | spectroradiometer[f] UVB-1, YES[g] | Oct 2005–Feb 2007 | 32.5 %[h] 69.3 % |
| | | EDD | spectroradiometer | Oct 2005–Jul 2006 | 17.1 % |
| | Briançon, France | EDD | spectroradiometer | Oct 2004–Sep 2005 | 7.9 % |
| (Ialongo, Casale, and Siani 2008) | Rome, Italy | EDR (noon) | Brewer MKIV UVB-1, YES | Sep 2004–Jul 2006 | 33 %[i] 30 % |
| (Tanskanen et al. 2007)[j] | 17 sites | EDD | 18 instruments | Sep 2004–Mar 2006 | up to 50 %[k] |
| (Bernhard et al. 2015)[l] | 13 stations | EDD | 13 instruments | Sep 2004–Dec 2012 | −1 % to 24 %[m] |
| (Weihs et al. 2008)[n] | Vienna, Austria | UV index (op) | Biometer | May–Jul 2007 | −10 % to 50 %[o] |
| (Janjai et al. 2014)[p] | Thailand | UV index (op) | Multi-channel UV radiometer | 2008–2010 | 43.6 %, 43.5 %, 28.7 %, 21.9 %[q] |



[a]Spectral represents the OMI spectral irradiance data, EDR is the erythemal dose rate and EDD is the erythemally weighted
daily dose. Op corresponds to the OMI data at its overpass time while noon means the data at local solar noon time.
[b]The validation statistic shown here is the bias with each study using slight different ways of calculation.
[c]The bias here is calculated as the median (OMI/Ground – 1) * 100.
[d]The bias is calculated as $100 \cdot \frac{1}{N} \sum_{i=1}^{N} \frac{OMI-Ground}{OMI}$, where N is the total number of data points.
[e]The bias is calculated as the mean (OMI – Ground)/Ground *100.
[f]The spectroradiometer used here is thermally regulated Jobin Yvon H10 double monochromators.
[g]The broadband UVB-1 is from Yankee Environmental System (YES).
[h]The bias is calculated as $100 \cdot \frac{1}{N} \sum_{i=1}^{N} \frac{OMI-Ground}{Ground}$, where N is the total number of data points.
[i]Same as [h].
[j]This study evaluated OMI surface EDD at 17 ground sites representing different latitudes, elevations and climate conditions
with 18 instruments, which include single and double Brewer spectrophotometers, NIWA UV Spectrometer Systems,
DILOR XY50 spectrometer, and SUV spectroradiometers. More detailed information can be found in this study.
[k]The bias is calculated same as [c]. For sites significantly affected by absorbing aerosols or trace gases, the bias can be up to 50

15  %.

[l]This study evaluated OMI EDD at 13 ground stations located throughout the Arctic and Scandinavia from 60° to 83º N. The
instruments installed include single-monochromator Brewer spectrophotometer, GUV-541 and GUV-511 multi-filter
radiometers from Biospherical Instrument Inc. (BSI).
[m]Same as [c].
[n]This study evaluated OMI UV index at 6 ground stations in the city of Vienna, Austria, and its surroundings. 6 Biometers
(Model 501, Solar Light) were used.
[o]The bias is calculated as (OMI/Ground – 1) * 100 and here shown is the result for clear-sky conditions.
[p]This study evaluated OMI UV index at four tropical sites in Thailand with each site having different time periods of data
between 2008–2010. The ground instrument installed is a multi-channel UV radiometer (GUV-2511) manufactured by BSI.
[q]The bias is calculated as [h], representing the four sites, respectively.



1    **Table 2. OMI data products and validation statistics used in the current study.**

| | Full name | Acronym | Unit |
|---|---|---|---|
| Data products | Full-sky overpass time erythemal dose rate | OP_FS EDR | mW m$^{-2}$ |
| | Full-sky solar noon erythemal dose rate | Noon_FS EDR | mW m$^{-2}$ |
| Validation statistics | Mean bias | MB | mW m$^{-2}$ |
| | Normalized mean bias | NMB | % |
| | Root-mean-square error | RMSE | mW m$^{-2}$ |
| | Root-mean-square difference | RMSD | mW m$^{-2}$ |
| | Normalized standard deviation | NSD | unitless |



1    **Table 3. The 31 ground observational sites from UVMRP and their geographical information.**

| Station ID | Location | Latitude (°N) | Longitude (°W) | Elevation (m) |
|---|---|---|---|---|
| AZ01 | Flagstaff, AZ | 36.06 | 112.18 | 2073 |
| CA01 | Davis, CA | 38.53 | 121.78 | 18 |
| CA21 | Holtville, CA | 32.81 | 115.45 | -18 |
| CO01 | Nunn, CO | 40.81 | 104.76 | 1641 |
| CO11 | Steamboat Springs, CO | 40.46 | 106.74 | 3220 |
| CO41 | Lamar, CO | 38.07 | 102.62 | 1131 |
| FL01 | Homestead, FL | 25.39 | 80.68 | 0 |
| GA01 | Griffin, GA | 33.18 | 84.41 | 267 |
| IL01 | Bondville, IL | 40.05 | 88.37 | 213 |
| IN01 | West Lafayette, IN | 40.47 | 86.99 | 216 |
| LA01 | Baton Rouge, LA | 30.36 | 91.17 | 6 |
| MD01 | Queenstown, MD | 38.92 | 76.15 | 5 |
| MD11 | Beltsville, MD | 39.01 | 76.95 | 64 |
| ME11 | Presque Isle, ME | 46.70 | 68.04 | 155 |
| MI01 | Pellston, MI | 45.56 | 84.68 | 230 |
| MN01 | Grand Rapids, MN | 47.18 | 93.53 | 424 |
| MT01 | Poplar, MT | 48.31 | 105.10 | 634 |
| MS01 | Starkville, MS | 33.47 | 88.78 | 88 |
| NC01 | Raleigh, NC | 35.73 | 78.68 | 120 |
| ND01 | Fargo, ND | 46.90 | 96.81 | 275 |
| NE01 | Mead, NE | 41.15 | 96.49 | 355 |
| NM01 | Las Cruces, NM | 32.62 | 106.74 | 1317 |
| NY01 | Geneva, NY | 42.88 | 77.03 | 219 |
| OK01 | Billings, OK | 36.60 | 97.49 | 317 |
| ON01 | Toronto, ON | 43.78 | 79.47 | 210 |
| TX21 | Seguin, TX | 29.57 | 97.98 | 172 |
| TX41 | Houston, TX | 29.72 | 95.34 | 76 |
| UT01 | Logan, UT | 41.67 | 111.89 | 1369 |
| VT01 | Burlington, VT | 44.53 | 72.87 | 390 |
| WA01 | Pullman, WA | 46.76 | 117.19 | 805 |



| WI01 | Dancy, WI | 44.71 | 89.77 | 381 |
|------|-----------|-------|-------|-----|



**Table 4. Regression statistics and other validation statistics for evaluating OMI OP_FS EDR with 31 ground observational sites**
**using different spatial collocation distances and temporal averaging windows.**

| statistics[a] | D = 50[b] | | | | D = 25 | D = 10 |
|---|---|---|---|---|---|---|
| | 5 min[c] | 10 min | 30 min | 60 min | 5 min | 5 min |
| N | 100801 | 100824 | 100880 | 100938 | 67628 | 17479 |
| R | 0.88 | 0.88 | 0.90 | 0.91 | 0.87 | 0.87 |
| Slope | 0.8 | 0.82 | 0.85 | 0.88 | 0.81 | 0.83 |
| Intercept | 19.8 | 18.1 | 15.2 | 13.1 | 19.7 | 18.7 |
| MB | -4.1 | -4.0 | -3.4 | -1.1 | -3.7 | -2.4 |
| RMSE | 39.8 | 38.4 | 35.6 | 33.1 | 40.7 | 40.5 |

[a]N is the total number of data pairs between OMI and ground observation for 31 sites altogether. R, slope, and intercept are
the values obtained from the linear regression. MB and RMSE represent the mean bias and root-mean-square error as
calculated in Eq. (1) and (2), respectively.
[b]D = 50, 25, 10 are the spatial collocation distances (D = 50 km, 25 km, 10 km) between an OMI ground pixel center and a
ground observational site.
[c]5, 10, 30 and 60 are the temporal averaging windows ($\Delta T = \pm$ 5, 10, 30 and 60 minutes) around OMI overpass time.



**Table 5. Same as Table 4 but for evaluating OMI Noon_FS EDR.**

| statistics | D = 50 | | | | D = 25 | D = 10 |
|---|---|---|---|---|---|---|
| | 5 min | 10 min | 30 min | 60 min | 5 min | 5 min |
| N | 100696 | 100725 | 100773 | 100841 | 67530 | 17442 |
| R | 0.88 | 0.89 | 0.90 | 0.91 | 0.87 | 0.87 |
| Slope | 0.87 | 0.89 | 0.92 | 0.95 | 0.88 | 0.89 |
| Intercept | 25.6 | 23.7 | 20.9 | 18.8 | 25.4 | 24.8 |
| MB | 10.1 | 10.2 | 10.9 | 13.2 | 10.4 | 10.7 |
| RMSE | 42.2 | 40.8 | 38.7 | 37.5 | 43.3 | 43.2 |





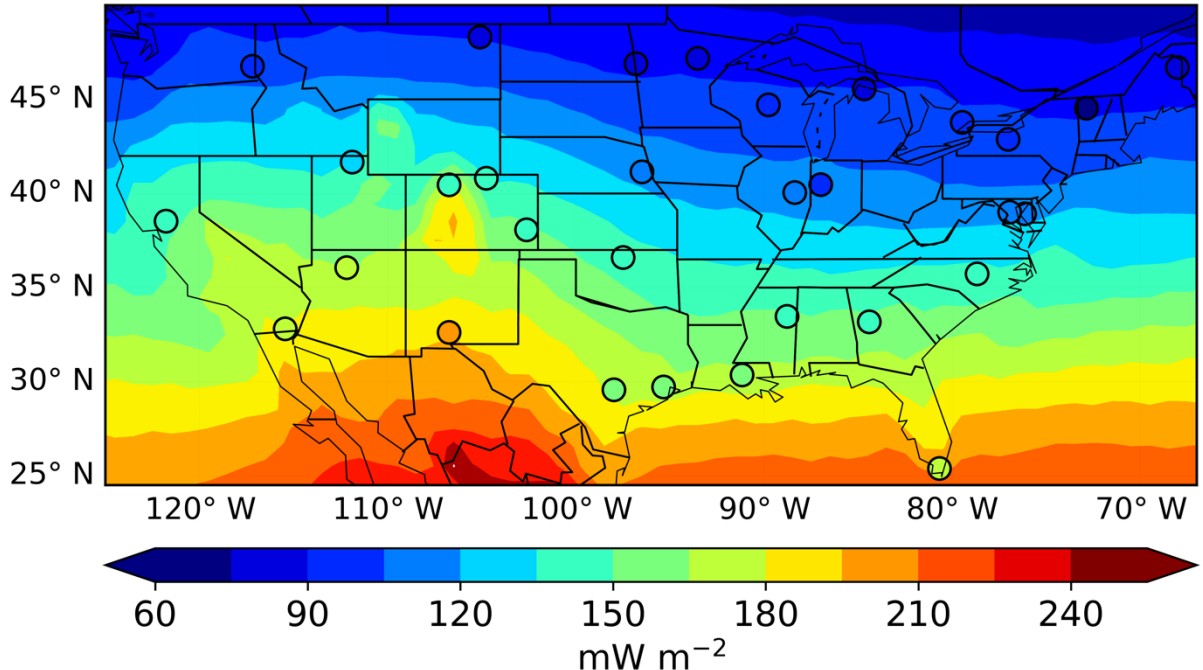

**Figure 1: Map of OMI level 3 EDR (mW m$^{-2}$) at solar noon time under full-sky conditions averaged over 2005–2017, overlaid with 31 ground observational sites averaged over 2005–2017 around solar noon time with ΔT = ±5 minutes.**







**Figure 2: Scatter plots of OMI EDR data with ground observations from year 2005 to 2017. (a) and (b) show the comparisons of OMI OP_FS and Noon_FS EDR with measurements at all of the 31 ground observational sites, respectively, while (c) and (d) only show the comparisons of OMI EDR with ground measurements at Homestead, Florida (FL01). In each scatter plot, also shown is the correlation coefficient (R), the root-mean-square error (RMSE), the number of collocated data points (N), the density of points (the color bar), the best-fit linear regression line (the dashed black line) and the 1:1 line (the solid black line).**





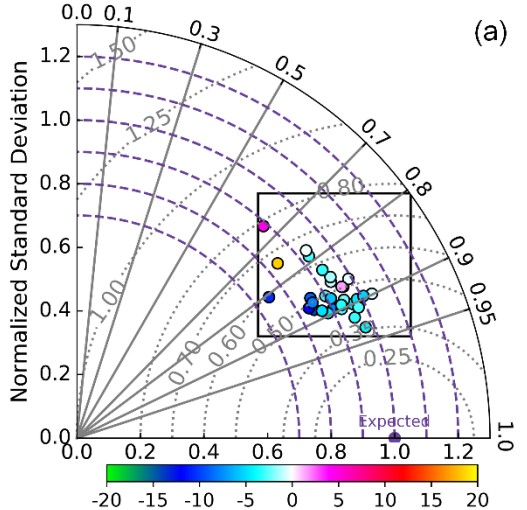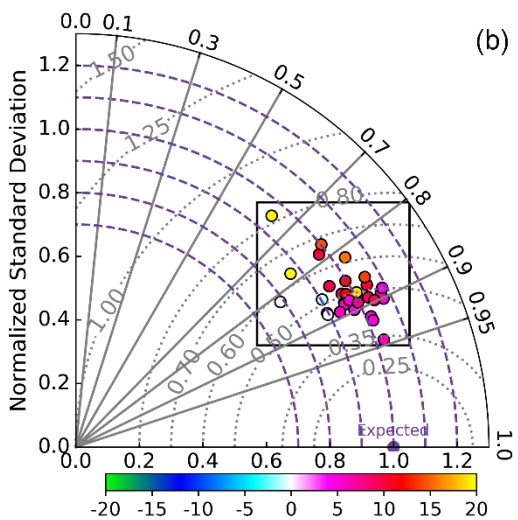

**Figure 3: Taylor Diagrams for evaluating OMI OP_FS EDR (a) and Noon_FS EDR (b) against 31 ground observational sites matched with D = 50 km and ΔT = ±5 minutes, respectively. The circles represent the ground sites and the color at each circle represents the NMB (%).**

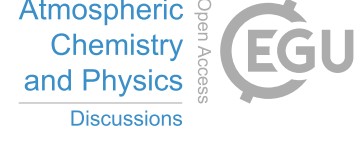

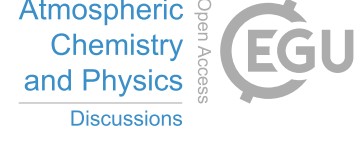

Figure 4: (a) and (b) are zoomed-in plots corresponding to the areas in the black box in Fig. 3(a) and Fig. 3(b), respectively. (c) is the zoomed-in plot for the evaluation of OMI OP_FS EDR with D = 50 km and ΔT = ±60 minutes against 31 ground sites. Sites denoted by squares in (a), (b) and (c) have NMB significant at 95% confidence levels. (d) shows the evaluation of OMI OP_FS EDR (triangles) and Noon_FS EDR (circles) with D = 50 km and ΔT = ±5, 10, 30 and 60 minutes against the ensemble of 31 ground observational sites.





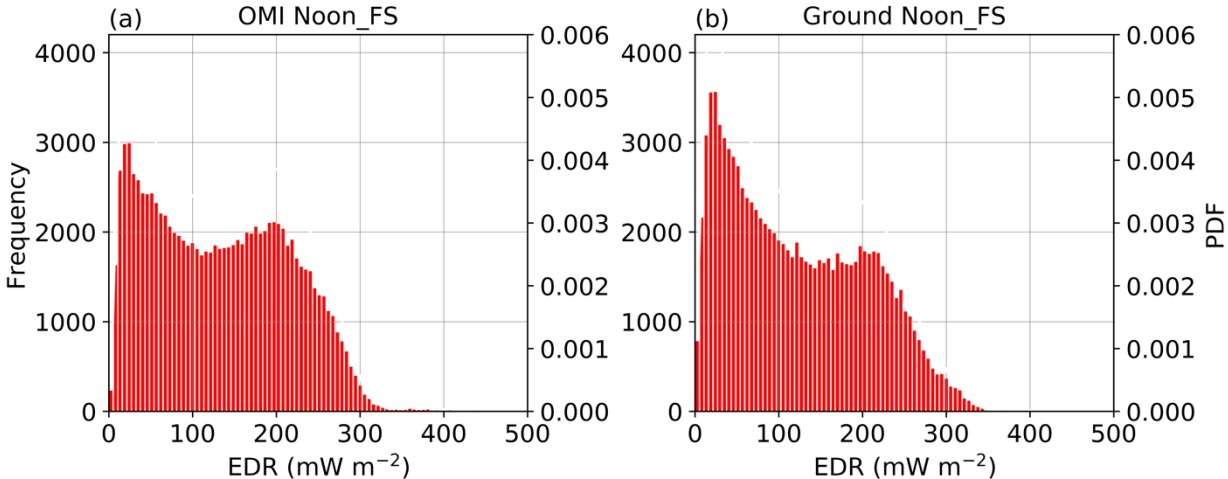

**Figure 5: Frequency (left axis) and PDF (right axis) of the surface EDR at the solar noon time for OMI (a) and 31 ground observational sites (b) for year 2005–2017. All the data pairs are matched with D = 50 km and ΔT = ±5 minutes.**





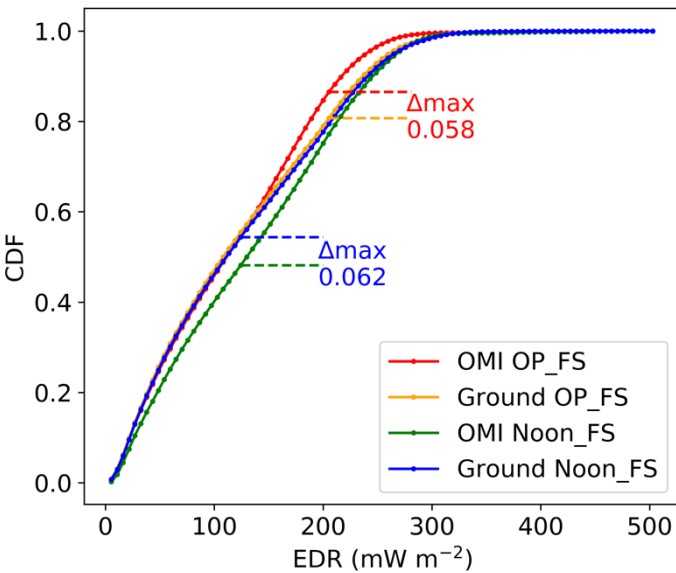

**Figure 6: Cumulative distribution functions (CDFs) of surface EDR from both OMI and 31 ground observational sites over 2005–2017. The maximum differences between OMI and ground observational CDFs are shown in the horizontal dashed lines and their values are shown as the labels.**



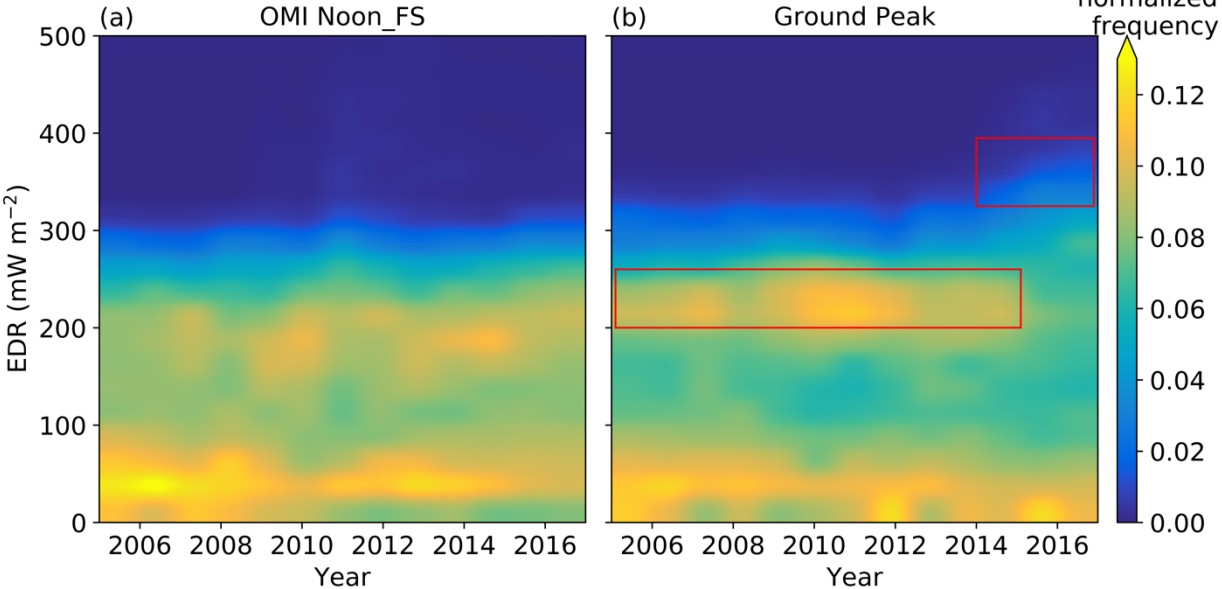

**Figure 7: Contour plot of normalized frequency of surface EDR from OMI Noon_FS (a) and ground peak (b) for 31 ground sites.
The ground peak refers to the highest dose rate found in a day at each site. The normalized frequency is calculated as follows:
first, the surface EDR from both OMI and ground observation are binned by 25 mW m$^{-2}$ for each year and then normalized by the
total number of data points for each year. A smooth effect at the contour line was also performed. The red box on the top in (b)
marks the areas where the ground peak EDR started to increase after staying consistent from 2005 to 2014 and the red box on the
bottom shows the high frequency occurrence areas of ground peak EDR.**





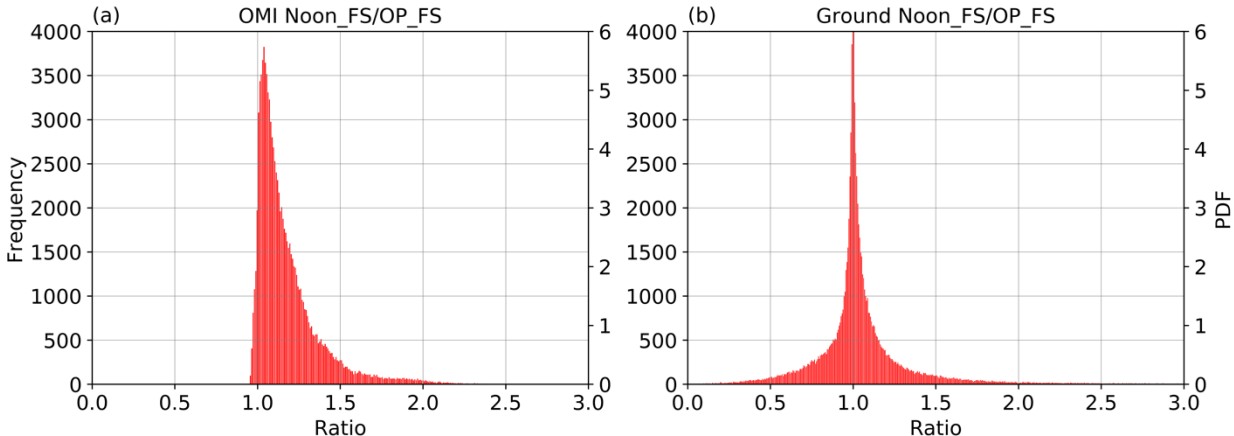

**Figure 8: Frequency (left axis) and PDF (right axis) of the EDR ratio of Noon_FS/OP_FS. (a) and (b) are for the OMI and ground ratio respectively. All the data pairs are matched with D = 50 km and ΔT = ±5 minutes for the 31 ground sites.**



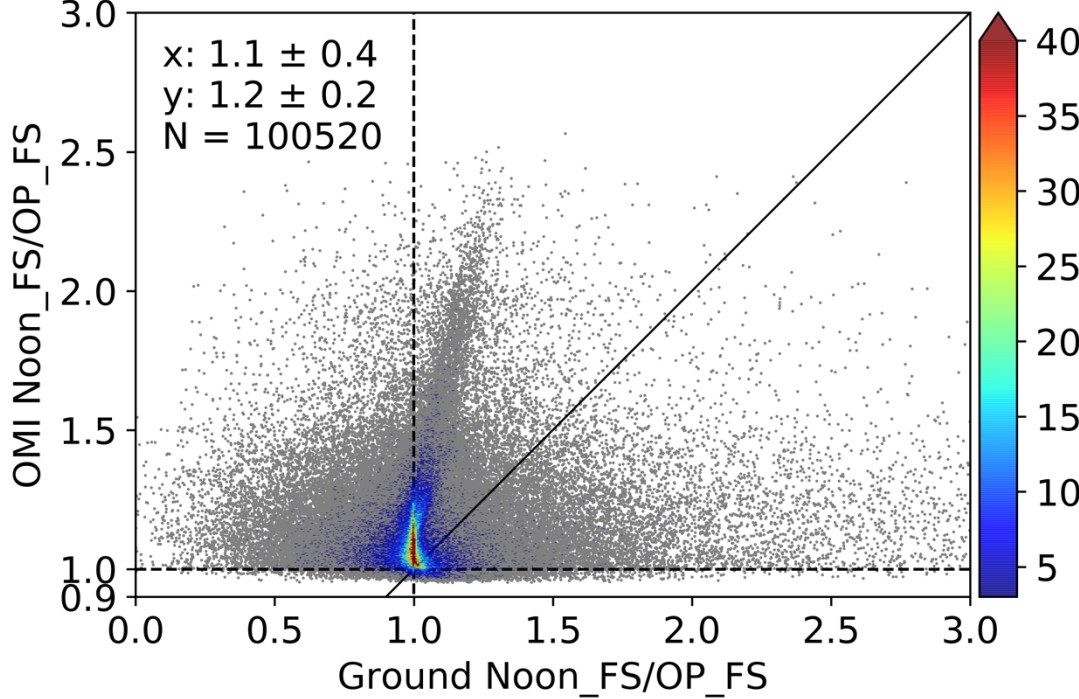

**Figure 9: Scatter plot of EDR ratio of Noon_FS/OP_FS from OMI and the ratio from ground measurements for 31 sites. All the**
**data pairs are matched with D = 50 km and ΔT = ±5 minutes. Also shown on the scatter plot is the number of collocated data**
**points (N), the density of points (the color bar), and the 1:1 line (the solid black line). Note the scale difference between x-axis and**
**y-axis.**





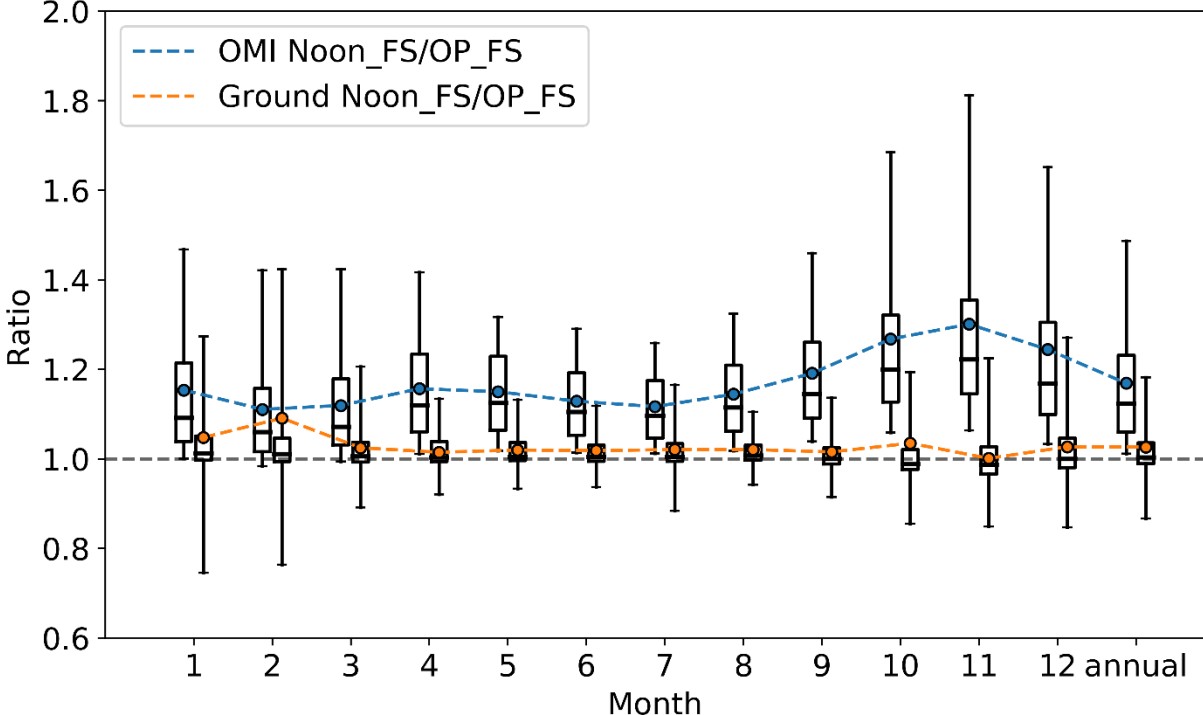

**Figure 10: Monthly EDR ratio of Noon_FS/OP_FS EDR from OMI (blue) and the ground (orange) for the 31 sites. The box-whisker plots show the 5th and 95th percentiles (whisker), the interquartile range (box), the median (black line) and the mean (the dots).**





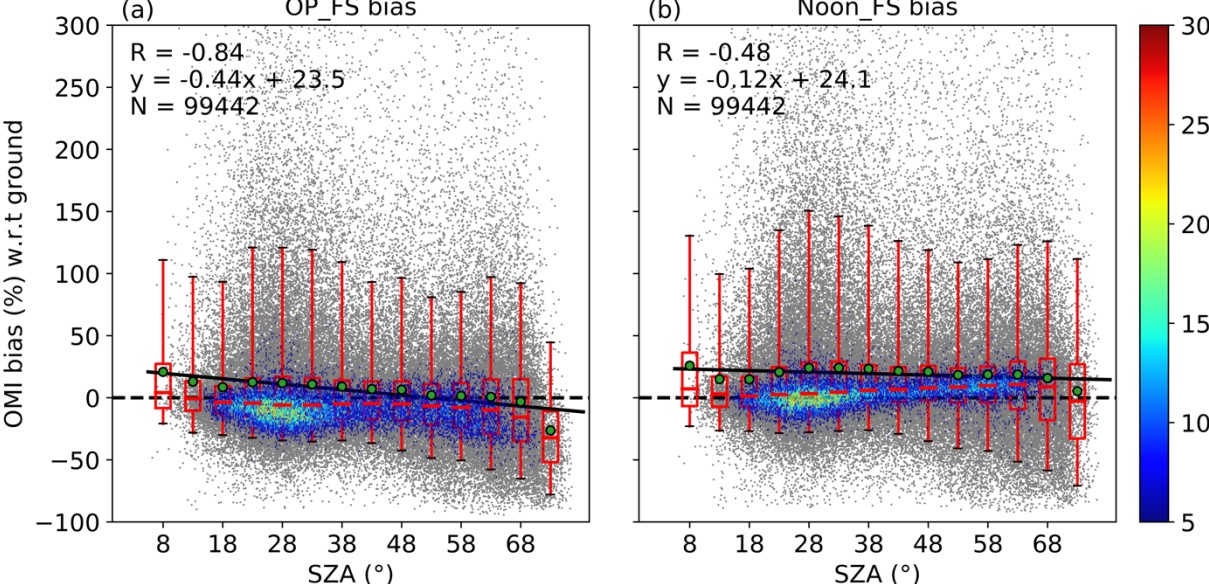

**Figure 11: Scatter plot of the relative bias (%) between OMI and ground observational EDR and the OMI overpass time SZA. (a) and (b) are for OP_FS and Noon_FS EDR comparison respectively. All the data pairs are matched with D = 50 km and ΔT = ±5 minutes for the 31 ground sites. The box-whisker plot of the bias is based on the binned SZA using a bin size of 5°. The box-whisker plots show the 5th and 95th percentiles (whisker), the interquartile range (box), the median (red line) and the mean (green dots). Also shown on the scatter plot is the number of collocated data points (N), the density of points (the color bar), the best-fit linear regression line (the solid black line), the regression equation and the correlation coefficient (R). Note that the linear regression is performed between the bin averaged bias and SZA.**





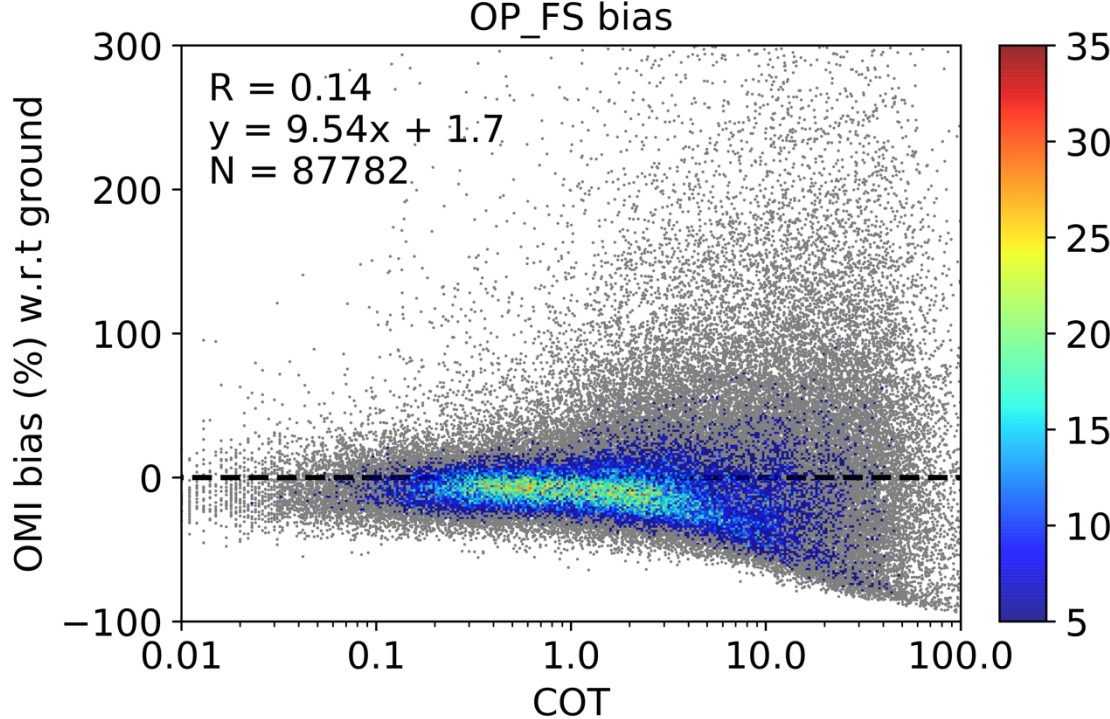

**Figure 12: Scatter plot of the relative bias (%) between OMI OP_FS and ground observational EDR and the OMI retrieved COT**
**(360 nm) for the 31 ground sites. Also shown on the scatter plot is the number of collocated data points (N), the density of points**
**(the color bar), the regression equation and the correlation coefficient (R).**



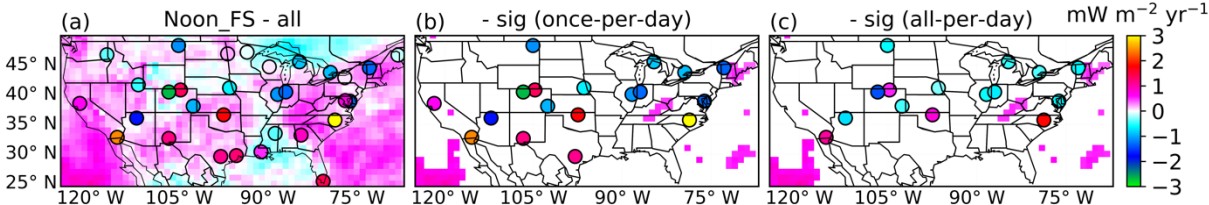

Figure 13: (a) is the distribution of the OMI level 3 solar noon time full-sky EDR trend over 2005–2017 overlaid with the trend at 31 ground observational sites calculated with D = 50 km and ΔT = ±5 minutes around local solar noon time. (b) is the same as (a) but only showing the areas and sites that are significant at the 95% confidence level. (c) shows the distribution of the trend at ground sites (significant at the 95% confidence level), computed with D = 50 km and temporally averaging all the data available in a day.





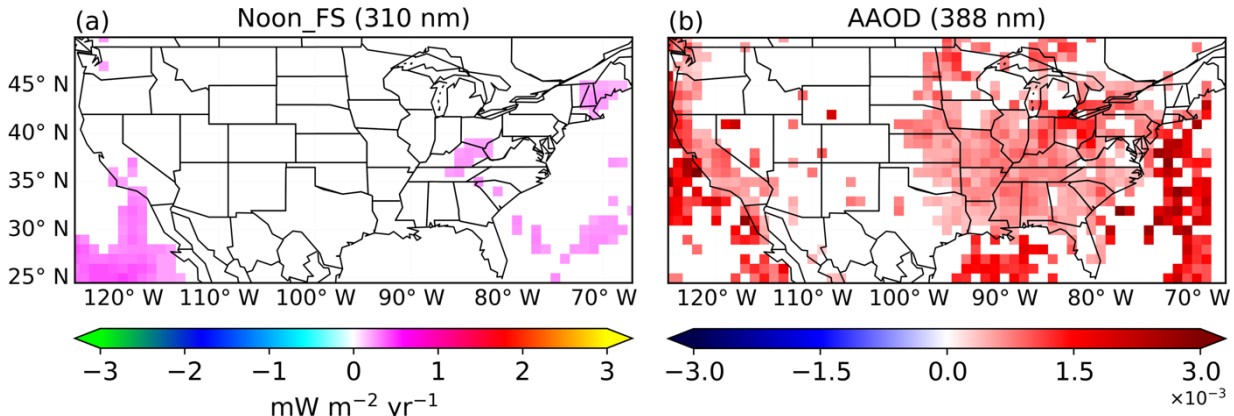

2 **Figure 14: Map of trend derived from OMI level 3 solar noon time full-sky EDR at 310 nm (a) and level 3 AAOD at 388 nm (b)**
3 **over 2005–2017. Shown are significant regions at the 95% confidence level.**



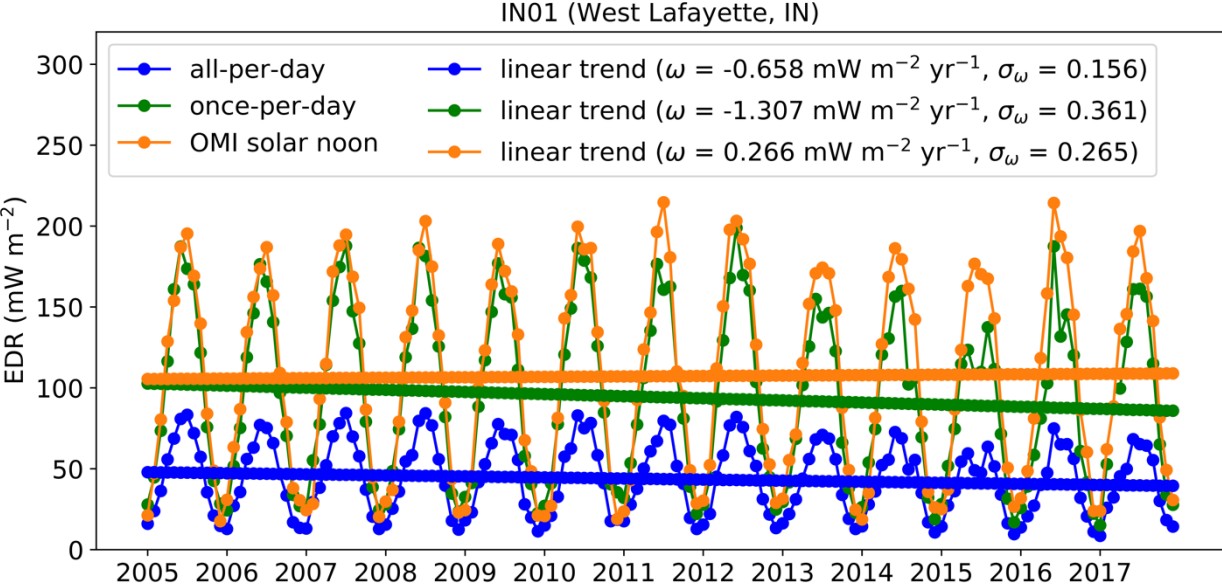

**Figure 15: Time series (dotted lines) of monthly OMI level 3 solar noon time full-sky EDR (orange) and ground observational EDR**
**from 2005–2017 using once-per-day (green) and all-per-day (blue) sampling method for site IN01. The once-per-day sampling**
**collects EDR data around local solar noon time while the all-per-day averages all the EDR data in a day. The straight lines are the**
**linear trends derived from Eq. (5).**

