# Peer review of "Surface erythemal UV irradiance in the continental United States derived from ground-based and OMI observations: quality assessment, trend analysis and sampling issues"

_Atmospheric Chemistry and Physics, 2018_

## Referee Comment (RC1) · Anonymous Referee #2 · 24 Sep 2018

The manuscript by Zhang et al. describes a comparison of erythemal dose rates (EDRs) measured by the space-borne Ozone Monitoring Instruments (OMI) and ground-based UVB1 pyranometers manufactured by Yankee Environmental Systems. The ground-based instruments are located in North America (30 sites in the U.S. plus one site in Canada) and are part of the UV-B Monitoring and Research Program (UVMRP) operated by the Colorado State University. This network is currently the largest ultraviolet (UV) radiation monitoring network operating in the U.S. and data from the network are therefore important for assessing the climatology of UV radiation in the U.S. A comprehensive comparison of the network's measurements with OMI observations has not been published to my knowledge. The subject of the paper is therefore

relevant for Atmospheric Chemistry and Physics. Unfortunately, I have concerns that the measurements of the ground-based instruments have not been processed correctly (see General Comments below) and this issue has to be resolved before the manuscript can be considered for publication in ACP.

General Comments

The manuscript compares ground- and satellite-based EDRs at the time of the satellite overpass and local solar noon. Figure 8 shows distributions of the ratio of measurements at noon and during the satellite overpass. For OMI data (Fig. 8a), the distribution is skewed towards values larger than one. This is the distribution that I would expect because OMI passes over the U.S. in the afternoon when the solar zenith angle (SZA) is larger than at noon. For days when atmospheric conditions stay constant between noon and the overpass, UV radiation is only controlled by the SZA and is therefore larger at noon than during the time of the overpass, resulting in a distribution like that shown in Fig. 8a. In contrast, the distribution of the ratio of noon and overpass measurements for ground-based data (Fig. 8b) is almost symmetric with a mean value of about 1. I find this result very surprising and it conflicts with my understanding of the radiative transfer at Earth's surface. I therefore suspect that the ground-based measurements were not processed correctly, and if so, this would have consequence for the majority of data presented in the manuscript. All affected data would have to be reprocessed.

Over the U.S., the OMI overpass occurs approximately between 0 and 2 hours after local solar noon. Hence, the SZA at local solar noon is almost always smaller than at the time of the satellite overpass. UV radiation must therefore be larger, on average, at noon than at the time of the overpass. Of course, UV radiation at the surface during the overpass time may occasionally be larger than during noon, for example, if enhancement by scattered clouds occurs. However it is highly unlikely that atmospheric conditions (e.g. clouds, aerosols, ozone) change in a systematic way between noon and overpass, resulting, on average, in higher absorption at noon to compensate for

the smaller SZA. For example, if clouds always had a larger optical depth at noon than at the overpass time, the effect could be explained. It is hardly possible that such a systematic cloud effect could occur for the majority of the 31 sites studied here. Hence, the symmetrical distribution centered at one (Fig. 8b) points to a problem in the data analysis (perhaps the calculation of the time of local solar noon is incorrect), which must be resolved before the paper can be published.

My second "major" comment refers to the selection of results being presented. The paper features an abundance of statistical analysis parameters but not the information that most readers would be most interested to see, i.e., the mean bias and range between ground-based measurements and OMI at each site. I therefore propose to include either a new table or a figure with box-whisker plots, which would show for each site the median and average relative bias, the interquartile range, and the 5th and 95th percentile range, for small SZA (e.g., 0 - 50 degree) and large SZA (50 - 75 degree), and for noon and overpass data. Such box-whisker plots would be similar to the box-whiskers shown in Figure 11, and would show data for each of the 31 sites separately, but only for two SZA ranges (which I think is sufficient). Such statistics are much more useful than Taylor diagrams, which are hard to interpret, and separating the results per site would allow to better discuss regional differences in the deviation of ground-based and OMI data, along with their causes (e.g., elevation, proximity to pollution and aerosol sources, clouds, etc.). If a figure with box-whisker plots is chosen, those could be spread-out over two or three rows to be able to show enough detail. Box-whiskers are easier to visually take-in than a table, but a table has the advantage that the numbers are defined. Finally, I don't see much value in Figures 5 and 6 and these could be removed (see details below).

The title of the paper should also mention the ground-based measurements. I propose: "Surface erythemal UV irradiance in the continental United States derived from ground-based and OMI observations: quality assessment, trend analysis, and sampling issues"

[Figure]

Specific comments:

P1, L17: As mentioned above, I find the conclusion in the following paragraph very surprising and physically impossible

"In addition, the ratio of EDR between solar noon to overpass time is often (95% in frequency) larger than 1 from OMI products; in contrast, this ratio from ground observation is shown to be normally distributed around 1. This contrast suggests that the current OMI surface UV algorithm would not fully represent the real atmosphere with the assumption of a constant atmospheric profile between noon and satellite overpass times."

The summary in the abstract needs to be revised once the issue of the surprising distribution of the ratio of noon and overpass ground-based data is resolved.

P1, L21: Change "Both OMI Noon_FS and ground peak EDR show a high frequency of occurrence of $\sim$ 20 mW m-2 over the period of 2005–2017. However, another high frequency of $\sim$ 200 mW m-2 occurs in OMI solar noon EDR while the ground peak values show the high frequency around 220 mW m-2, implying that the OMI solar noon time may not always represent the peak daily UV values." to: "The distributions of the OMI Noon_FS and the ground EDR were analyzed using data for the period 2005-2017. Both distributions have local maxima at about 20 mW m-2. The overall maximum of the distribution is 200 mW m-2 for the OMI and 220 mW m-2 for the ground-based data."

P2, L35: The references of the two assessment reports (UNEP, 2007; WMO, 2010) are now fairly old and should be updated by references to the latest assessment reports (UNEP 2015 and WMO 2014).

P2, L40: Bigelow et al., 1998; Sabburg et al., 2002; Levelt et al., 2006 is an odd collection of rather old papers. Perhaps some newer papers should be cited also.

L55: Explain acronyms UVMRP and USDA.

L56: The Brewer network is also still active in the U.S., see https://www.esrl.noaa.gov/gmd/grad/neubrew/ Please mention it!

L68: Also mention an example from South Americal, e.g., http://dx.doi.org/10.1016/j.jphotobiol.2012.06.013

L142: the standard definition for UV-B is 280-315 nm, not 280-320 nm.

L146-147: For satellite validation, the uncertainty for SZA > 80° is of lesser importance. What is the uncertainty for the SZA range applicable to OMI validation? (While the specification of deviations from measurements with the standard triad is interesting, the 2.8% specified here is not an uncertainty because measurements of the triad are not free from uncertainty.) Also, please specify the confidence interval of all uncertainty specifications.

L148: I don't understand what the authors want to emphasize with the sentence starting with "In spite of this," Do you mean that having small uncertainties is more important when quantifying geographical difference than for satellite validation? If my interpretation is correct, please explain why you think that is the case!

L175: I am not sure how to interpret "Correspondingly, the temporal mean of ground observation within Delta T is compared to the spatial mean of OMI data within D." As noted earlier, ground based data were aggregated into 3-minute averages. So there can only be 3 ground-based measurements in a +/- 5 min time period. Likewise, There are only one or two OMI measurements within 50 km from the ground site, based on the OMI pixel size discussed earlier. The sentence should be clarified.

L180: Specify whether the number of data pairs refers to all sites or each individual site.

L186-192: Please also provide the formulas for NSD and RMSD. In particular, I am not sure what the difference is between RMSE and RMSD. Equations would clarify this. It is also inconsistent to name the "normalized standard deviation" NSD but the "normalized

root-mean-square difference" RMSD. It should be NRMSD. Replace "shown in x and y axis respectively" with "shown both on the x and y axis". Replace "shown as the radius from the expected point" with "shown as concentric circles around the point labeled "expected" in Fig. 3a and 3b.

L222: n was already defined as the total number of data points on line 203. Don't use the same symbol for different quantities!

L243 and Fig. 2a, b: One would expect that ground-based and OMI measurements agree better at overpass time than noon. This is indeed the case (as described) for MB and RMSE. However, surprisingly, the regression line is closer to the (ideal) 1:1 relationship for Noon_FS data compared to OP_FS data. This observation should be mentioned in the text, and reasons should be given. Perhaps this observation is a result of errors in the processing of ground-based measurements discussed earlier.

Fig. 3 and associated text. I find it very strange that overpass data have a negative bias and noon data a positive one. This points again to a problem in the calculation of one (or both) of the datasets.

Figure 5 and associated text. While I understand how the figures were calculated, I don't understand why these distributions are important to show. The figures mix in data from all sites (with greatly different cloud and aerosol conditions) and seasons. The figure depends on many variables, which I can't decouple in my head when looking at these distributions, and therefore, I don't know what to learn from these distributions. While there are differences between the distributions for the ground and OMI measurements (which ideally shouldn't exist) I wouldn't be able to grasp from this figure what might have gone wrong with the UV retrieval from OMI data (assuming that the distribution for the ground-based data is correct). I would consider removing this figure and the text that goes with it.

Fig 6 and associated text. Again, I am not sure why this figure is important. I understand that it confirms that OMI surface EDR and ground-observed EDR were drawn

from the same distribution, but is this confirmation really important? What can be learned that is not already known from the correlation coefficient? Hence, I suggest to consider removing this figure also.

L286: define "peak UV", e.g., move or copy "peak refers to the highest dose rate found in a day at each site" from the caption of Fig. 7 to the text.

Fig 7 compares two different quantities and it is therefore not surprising that there are differences. For example, the observation of clouds within the OMI pixel will always lead to a reduction of UV radiation because cloud modification factors are <= 1. In contrast, ground-based measurements can be enhanced beyond the clear-sky value during broken cloud conditions. This is a well-known phenomenon, e.g., http://doi.org/10.1038/371291a0. It is therefore not surprising that the high-frequency band is at 200 mW m-2 for OMI Noon-FS data and 220 mW m-2 for "Ground Peak" data. Since it is the goal of the paper to validate OMI data, I see little value in comparing two different quantities and concluding that "OMI solar noon time EDR may not always represent the high peak value on a daily basis due to the varying atmospheric conditions." I suggest replacing the right panel of Fig. 7 with the distribution of Noon_FS data from the ground based measurements.

L321-325. Results discussed here and in Figure 8 do not make sense to me. Please see my General Comment earlier. For these reasons, I also disagree with the conclusion: "This indicates that the current OMI surface UV algorithm would not fully represent the real atmosphere with the assumption of constant atmospheric conditions being made and could thus induce errors in estimating surface UV irradiances." I suspect a problem with the processing of ground-based data, not of OMI.

Having said this, I note that Bernhard et al. (2015) (see: https://www.atmos-chem-phys.net/15/7391/2015/) have reported a problem in the conversion from OMI overpass EDRs to EDRs at local solar noon by the OMI UV algorithm. They conclude "Additional analysis suggests that the pattern is likely due to a systematic error of up to +/-0.5

degree in the calculation of the local-noon SZA by the [OMI] algorithm. For a SZA of 80 degree, a 0.5 degree error in SZA results in a UVI error of about 8 %". The authors should look at this publication and determined whether this problem also affects their data. According to my judgment, the problem is of less importance for the data of the USDA network than for data from the high-latitude sites discussed by Bernhard et al. (2015) because SZAs at the time of the satellite overpass are much smaller at lower latitudes.

L358: I believe that the larger noise in the bias at larger COT values has additional reasons than the one noted in the paper. For example, it is difficult from space to estimate the COT if the COT is large (e.g., > 10). In contrast, clouds with COT=100 attenuate UV radiation much more than those with COT=10.

Figure 13: It would be better to show trends per year or decade in percent instead of absolute values (in mW m-2 yr-1). The significant trend in one tiny part of California is almost not worth mentioning in the text (L 372).

Figure 14a: Also this plot should be presented in percent. (If you decide to keep trends in absolute terms, the unit should be changed to that of spectral irradiance: mW m-2 nm-1 yr-1.)

L377: I don't understand why Zhang et al. (2017) found a significant positive trends over the western U.S. using OMI AOD for 2005–2015 while such trend was not found by the same author in the present work. What is the difference? The fact that this study considers measurement between 2005 and 2017 instead of 2005 and 2015?

L380: Explain acronym "AAOD" I presume it means "absorption aerosol optical depth." Indicate that this parameter is part of the standard OMI data products. AAOD in the UV-B is very difficult to measure from space since most of the absorption is close to the ground. Indicate the uncertainty of OMI AAOD. Could the small AAOD trends (<0.003 per year) that are reported here be a result of drifts in OMI data?

L387: Monthly averages can be greatly biased if only 10 days are available, in particular if measurements occur only at the beginning or end of a month. It would be best to repeat the trend analysis using only months with 20 or more days of data, and compare the results to ensure that trend estimates are robust and not driven by missing days.

The conclusions need to be changed to reflect changes to the text resulting from my comments above. This applies in particular to lines 437-446 where the distributions of noon/overpass data are discussed.

L454-459: The suggestion that trends in UV seen in the ground measurements are caused by trends in AAOD are highly speculative, and this should be stated. I believe that trends in AAOD derived from OMI measurements at 310 nm are rather uncertain. Furthermore, assessments of trends in AOD reported in the paper are only based on OMI (lines 377-379). The conclusions that there are no trends in AOD is premature because of the difficulty to probe the lower troposphere from space in the UV-B. It would be good to check whether the ground-based AERONET sunphotometer network has reported trends in AOD and AAOD for the period of the paper and to use those data to interpret trends in EDRs (although AERONET also does not have wavelengths in the UV-B).

Technical Corrections

General: It would be helpful to define acronyms for "overpass time EDR" and "local solar noon EDR" at the beginning and then use these acronyms in the text instead of repeating the same phrases.

P1, L13: Change "OMI data overall has ∼4% underestimate for overpass EDR while ∼8% overestimate for the solar noon time EDR" to "OMI underestimates the overpass EDR by about 4% on average and overestimates the solar noon time EDR by 8%."

P1, L17: "with SZA" > "for SZA"

P1, L20: "viability" > "variability"

[Figure]

P1, L21: Explain "Noon_FS"

L44: "in many locations" > "at many locations

L47: "In addition, the surface UV irradiance, denoted as 'erythmal weighted'" > "In addition, the erythemally weighted irradiance" (spectral irradiance is also surface UV irradiance in this context!)

L49: "the incoming solar radiation" > "the solar irradiance"; and "according to" > "with"

L51: "derive UV index" > "derive the UV index"

L52: "UV index" > "the UV index"

L60: "has been" > "have been" (data is plural)

L61: "the OMI data has" > "OMI data have"

L64: "in many sites" > "at many sites"

L100: "tables are the" > "tables is the"

L119: "triangular slit function with full width half maximum" > "a triangular slit function with full width at half maximum"

L177: delete "different"

L187: "normalized room-mean-square" > "normalized root-mean-square" (not "room"!)

L262 "larger" > "large"

L364: "range principally controlled" > "range that is greatly affected"

L366: "identified trend of surface EDR" > "trends in surface EDR"

L388: "In contrast, ground observation shows" > "In contrast to trends derived from OMI data, ground observations show"

L425: Explain acronyms "TEMPO and GEMS"

---

## Referee Comment (RC2) · Anonymous Referee #1 · 30 Sep 2018

This paper deals with the validation of the UV irradiance retrieved from OMI against ground-based measurements at 31 sites in the US. It is a well written study which with the following corrections it could be published in the Atmospheric Chemistry and Physics journal.

My most serious comment has to do with the collocation logic of satellite and ground-based measurements in terms of atmospheric conditions persistence. I do not see the point at comparing 31 ground stations with satellite retrievals and making such a big assumption. The comparison at local noon seems to not follow scientific criteria because the transferability of the sensitive UV irradiance at different time positions

undermines fundamental assumptions, but, at the same time, has an interpretation oriented to usefulness and applicability as at local noon the impact of UV irradiance on human health is major, so I believe that the whole effort is worthwhile. To my understanding there are no alternative ways to compare these datasets at local noon but it is highly recommended to include a description on the error percentages that were added by each one of the atmospheric parameters that has impact the UV irradiance. As a result, my suggestion to the authors is to include an analytical and quantified list of the atmospheric parameters that affects this time transferability of UV irradiance observations and discuss the magnitude of these impacts.

On page 13, line 387, the trend analysis needs to be performed with reliable monthly averages, so filter the data as to represent mean values with minimum missing data (e.g. at least 20 days of data per month).

I recommend also the authors to renew the reference list throughout the paper with updated and more recent results.

At line 20 of the Abstract maybe the authors want to say "variability" instead of "viability"?

At line 187 change the last letter of the word "room" with "t".

The RMSD is the same magnitude as the RMSE, so for the normalized RMSD use NRMSE or change the RMSE to RMSD as to not confuse the reader.

At line 380 what is the AAOD? Add a description before using any abbreviation.

At line 425 What are the "TEMPO and GEMS"? Add a nomenclature and abbreviation table for the whole document.

In Figure 5 change the x axis from 0 to 400.

In Figures 6 and 15 add grid lines.

In Figure 8 change the x axis borders from 0.0 to 2.5 (now its until 3.0). Add also some

in plot statistics (e.g. median or mean) and discuss in detail the form of the different distributions of plots a and b.

In Figure 10 add horizontal grid lines.

Finally, in Figure 11 place plot b bellow plot a and increase the aspect ratio as to cover the whole width. Then, at x axis place the SZA steps every 4-5 degrees as to describe better these interesting plots.

The overall analysis merits publication and is able to forward the use of Earth Observation techniques in order to measure or estimate with high accuracy the UV irradiance levels. I strongly believe that after the above revisions the paper could be published in the Atmospheric Chemistry and Physics journal.

---

## Referee Comment (RC3) · Anonymous Referee #3 · 8 Oct 2018

The manuscript by Zhang et al., "OMI surface UV irradiance in the continental United States: quality assessment, trend analysis, and sampling issues", presents a study to evaluate the estimates of surface UV from OMI satellite measurements against ground-based measurements and additionally carries out a trend analysis. There have been several validation studies of OMI surface UV before, unfortunately it remained unclear what is the understanding that this manuscript adds to the current knowledge.

Nothing was said about the quality and uncertainties of the ground-based measurements. And since the results were peculiar enough, it was not clear whether the ground measurements tell something about OMI UV performance or whether it is other way

around: comparison against OMI UV tells something about the systematic errors in the ground-based measurements. In my opinion, it takes a major revision before the manuscript can be fully evaluated. I will explain and clarify these points below.

Main comments:

As the authors quite correctly mention, the information about the atmospheric conditions comes from the OMI measurement at the overpass time. So this statement is already suggesting that only the overpass time comparisons are meaningful. And indeed, my suggestion is to exclude the local noon time comparisons. However, since the authors presented also the local noon time comparisons with strange results, I want to discuss these results briefly below.

The SZA is always somewhat lower at the local noon time than at the overpass time, resulting in larger UV irradiance at the local noon by this SZA effect. This means that there has to be some very clear and systematic effect in cloudiness to compensate this, if the results in the Figure 8b are true. Cloud effect would the only plausible explanation and it would mean that at the OMI overpass time there should be systematically (on average) lower cloud amount than at the local solar noon over the stations studied. From the results of Meskhidze et al. 2009, the opposite diurnal cloud effect seems to be the case, on average thicker (higher cloud optical depth) clouds in the afternoon than before (at Terra overpass time) over most of the continental US. The overpass time of Terra is before noon, but I think this should give an indication, at least, about the sign of the systematic difference between noon and OMI overpass. So how do you explain your Figure 8b? It is absolutely necessary to include ancillary data or publication citation to convince the reader about the behavior in the Figure 8b. Otherwise, he/she is left with an impression that something has to be wrong with the ground-based measurements.

If the results in the Figure 8b are not understood and explained, the reader can only assume that there has to be some angular dependent error left in the measurements

to cause this. And indeed, there was no single word about the uncertainty of the ground-based measurements. The reference to Lantz et al. 1999 was just given, without any further discussion. So please explain in detail the corrections applied. The correction should be ozone and SZA dependent, as they discuss in Lantz et al. 1999. From where you take the ozone values for the correction? How often the reference is calibrated? How often these instruments participate in inter-comparison campaigns? I think it would be an informative plot to show also the typical correction factor, plotted as a function of SZA, for two very different ozone amounts. The angular calibration is also a function of cloudiness, so please discuss in detail how it is included in the correction of ground-based measurements.

Currently the trend analysis part does not offer anything consistent. For the careful reader, the main message seems to be that the ground-based measurements result in both negative and positive trends, no matter how the data are selected, and even for stations that are almost side by side. So perhaps this trend analysis could be also excluded, or the authors explain what is the consistent message it brings. For instance, let's take a look at the East coast of US, some sites give slightly negative trend (13b), the southern most shows a positive trend. If one studies Zhang et al. 2017 in detail, it is clear that there is no AAOD trend in this region, while there is a negative trend in AOD (from OMI, but also from other instruments). So one would assume slight positive trend in surface UV, but in the contrary there is negative trend in two sites.

The changes in AAOD alone cannot explain these trends, so it is absolutely important to consider the simultaneous changes in AOD as well. If one selects those regions from Zhang et al. 2017 where both AOD and AAOD (from OMI) shows positive change, then the most probable sites are Holtville, CA and Las Cruces, NM, where based on this change, one would assume to see negative change in surface UV. However, both stations show positive change. These are just two examples why I argue that in its current form, the manuscript fails to offer consistent and convincing message about the trend analysis. My two main comments might be linked: if the quality of the groundbased measurements is not sufficient and/or properly considered, then these issues might become visible both in the results shown in Figure 8b and also in the trend analysis. It can be also, that in any case, the signal is too weak to detect any meaningful trend, but if so, then the discussion and comparison against OMI AAOD is not justified.

Specific comments:

Line 364, you mention that the absorbing aerosols could be the reason for the OMI UV trend. It is not the likely reason, since the correction is taken from monthly climatology. So what did you mean?

Line 370, there is a better reference to Kinne et al. (Kinne et al. 2013 below).

Line 361, if you include 310nm, perhaps comparison to the 380nm trend would bring something useful, since it does not have any significant ozone absorption.

Table 3. If you used ordinary least squares for regression, please remember that it gives a systematically biased slope when there is uncertainty in x-axis. These numbers, slopes in particular, would be informative if the method for the regression is correct one. See Cantrell et al. 2009 or Pitkanen et al. 2016. Please explain the method that was used and the possible limitations.

REFERENCES:

Cantrell, C. A.: Technical Note: Review of methods for linear least-squares fitting of data and application to atmospheric chemistry problems, Atmos. Chem. Phys., 8, 5477-5487, https://doi.org/10.5194/acp-8-5477-2008, 2008.

Kinne, S., D. O'Donnel, P. Stier, S. Kloster, K. Zhang, H. Schmidt, S. Rast, M. Giorgetta, T. F. Eck, and B. Stevens, MAC-v1: A new global aerosol climatology for climate studies, J. Adv. Model. Earth Syst., 5, 704–740, doi: 10.1002/jame.20035, 2013.

Meskhidze, N., Remer, L. A., Platnick, S., Negrón Juárez, R., Lichtenberger, A. M., and Aiyyer, A. R.: Exploring the differences in cloud properties observed

by the Terra and Aqua MODIS Sensors, Atmos. Chem. Phys., 9, 3461-3475, https://doi.org/10.5194/acp-9-3461-2009, 2009.

Pitkänen, M. R. A., S. Mikkonen, K. E. J. Lehtinen, A. Lipponen, and A. Arola, Artificial bias typically neglected in comparisons of uncertain atmospheric data, Geophys. Res. Lett., 43, 10,003–10,011, doi: 10.1002/2016GL070852, 2016.

---

## Author Comment (AC1) · 23 Dec 2018

We sincerely thank the editor and reviewers for taking the time to review our manuscript and providing constructive feedback to improve our manuscript. We have revised the manuscript accordingly by following the reviewers' suggestion. Below shown are the original comments from reviewers in black and our corresponding responses in blue.

**Comments by RC1:**

This paper deals with the validation of the UV irradiance retrieved from OMI against ground-based measurements at 31 sites in the US. It is a well written study which with the following corrections it could be published in the Atmospheric Chemistry and Physics journal.

My most serious comment has to do with the collocation logic of satellite and ground- based measurements in terms of atmospheric conditions persistence. I do not see the point at comparing 31 ground stations with satellite retrievals and making such a big assumption. The comparison at local noon seems to not follow scientific criteria because the transferability of the sensitive UV irradiance at different time positions undermines fundamental assumptions, but, at the same time, has an interpretation oriented to usefulness and applicability as at local noon the impact of UV irradiance on human health is major, so I believe that the whole effort is worthwhile. To my understanding there are no alternative ways to compare these datasets at local noon but it is highly recommended to include a description on the error percentages that were added by each one of the atmospheric parameters that has impact the UV irradiance. As a result, my suggestion to the authors is to include an analytical and quantified list of the atmospheric parameters that affects this time transferability of UV irradiance observations and discuss the magnitude of these impacts.

Response: We are very thankful for the reviewer's overall comments on the manuscript and the suggestions to help us improve the manuscript. Regarding the suggestion to provide an analytical and quantified list of the atmospheric parameters that affects this time transferability of UV irradiance observations and their respective impacts, we understand that it is important to quantify the individual effect. However, it is difficult to quantify the impact from each atmosphere parameter due to the following reasons. The erythemally weighted irradiance covers the range from 300 to 400 nm, for which using this weighted irradiance information alone would not be able to attribute the bias to a specific cause such as changes in ozone or aerosols. Additional information such as the comparison of OMI spectral irradiance with ground measurements would be helpful in revealing the cause, which is beyond the scope of our current study. Nevertheless, in our current work, we provide the overall combined effects from different parameters, and we also further outlined and studied possible factors (such as cloud, aerosols, and ozone). Unfortunately, the measurements of these factors from space also have their uncertainties. In addition, we added some discussion on the limitation of our current work in this perspective in Sect. 5 (Conclusion and Discussion). Furthermore, clarification is made regarding the inter-comparison between OMI and ground-based observations at local solar noon; especially we now use the local solar zenith angle (instead of local hour) to define the local noon time that is consistent with the OMI algorithm itself. The results are improved and figures are updated.

Overall, the main part of the paper is the long-term evaluation of the OMI product to reveal if there is any systematic bias and trend in its surface UV product, and if the polar-orbiting satellite with once-per-day sampling is sufficient to decrilbe the climatology of surface EDR, especially daily maxima of EDR that is of high interest for studying the

impact of UV exposure on the health of biosphere and human. To some extent, this work is in analogy with the trend analysis of surface temperature first before we can figure out the causes for the trend (if there is any). For the bulk part, we didn't find any significant and coherent trend in surface UV, $O_3$, aerosols, and clouds from OMI.

On page 13, line 387, the trend analysis needs to be performed with reliable monthly averages, so filter the data as to represent mean values with minimum missing data (e.g. at least 20 days of data permonth).
Response: We have followed the reviewer's suggestion to calculate the monthly average only for months with at least 20 days of data available.

I recommend also the authors to renew the reference list throughout the paper with updated and more recent results.
Response: We agree with the reviewer. We have added a few newer references when appropriate to make our work more relevant.

At line 20 of the Abstract maybe the authors want to say "variability" instead of "viability"?
Response: Corrected

At line 187 change the last letter of the word "room" with "t".
Response: Corrected.

The RMSD is the same magnitude as the RMSE, so for the normalized RMSD use NRMSE or change the RMSE to RMSD as to not confuse the reader.
Response: Thanks for the suggestion and we have updated this in the manuscript (Eq. (4)).

At line 380 what is the AAOD? Add a description before using any abbreviation.
Response: We have described the AAOD in the earlier Sect. 2.1 and we apologize for the many abbreviation and acronyms used in the current manuscript. We have modified Table 2 to better organize the abbreviation we have used in the current manuscript.

At line 425 What are the "TEMPO and GEMS"? Add a nomenclature and abbreviation table for the whole document.
Response: Corrected.

In Figure 5 change the x axis from 0 to 400.
Response: Corrected.

In Figures 6 and 15 add grid lines.
Response: Corrected.

In Figure 8 change the x axis borders from 0.0 to 2.5 (now its until 3.0). Add also some in plot statistics (e.g. median or mean) and discuss in detail the form of the different distributions of plots a and b.

Response: Thanks for the suggestion. We have replotted the figure and discussed the figure in more details in the text.

In Figure 10 add horizontal grid lines.

Response: Corrected.

Finally, in Figure 11 place plot b bellow plot a and increase the aspect ratio as to cover the whole width. Then, at x axis place the SZA steps every 4-5 degrees as to describe better these interesting plots.

Response: Thank you for the suggestion and we have replotted the figure based on the suggestion.

The overall analysis merits publication and is able to forward the use of Earth Observation techniques in order to measure or estimate with high accuracy the UV irradiance levels. I strongly believe that after the above revisions the paper could be published in the Atmospheric Chemistry and Physics journal.

We thank the reviewer again for the positive feedback.

---

## Author Comment (AC2) · 23 Dec 2018

We sincerely thank the editor and reviewers for taking the time to review our manuscript and providing constructive feedback to improve our manuscript. We have revised the manuscript accordingly by following the reviewers' suggestion. Below shown are the original comments from reviewers in black and our corresponding responses in blue.

**Comments by RC2:**

The manuscript by Zhang et al. describes a comparison of erythemal dose rates (EDRs) measured by the space-borne Ozone Monitoring Instruments (OMI) and ground-based UVB1 pyranometers manufactured by Yankee Environmental Systems. The ground-based instruments are located in North America (30 sites in the U.S. plus one site in Canada) and are part of the UV-B Monitoring and Research Program (UVMRP) operated by the Colorado State University. This network is currently the largest ultraviolet (UV) radiation monitoring network operating in the U.S. and data from the network are therefore important for assessing the climatology of UV radiation in the U.S. A comprehensive comparison of the network's measurements with OMI observations has not been published to my knowledge. The subject of the paper is therefore relevant for Atmospheric Chemistry and Physics. Unfortunately, I have concerns that the measurements of the ground-based instruments have not been processed correctly (see General Comments below) and this issue has to be resolved before the manuscript can be considered for publication in ACP.

Response: We are very grateful for the reviewer's thoughtful and detailed comments to improve our manuscript. We have addressed this concern in the General Comments.

**General Comments**

The manuscript compares ground- and satellite-based EDRs at the time of the satellite overpass and local solar noon. Figure 8 shows distributions of the ratio of measurements at noon and during the satellite overpass. For OMI data (Fig. 8a), the distribution is skewed towards values larger than one. This is the distribution that I would expect because OMI passes over the U.S. in the afternoon when the solar zenith angle (SZA) is larger than at noon. For days when atmospheric conditions stay constant between noon and the overpass, UV radiation is only controlled by the SZA and is therefore larger at noon than during the time of the overpass, resulting in a distribution like that shown in Fig. 8a. In contrast, the distribution of the ratio of noon and overpass measurements for ground-based data (Fig. 8b) is almost symmetric with a mean value of about 1. I find this result very surprising and it conflicts with my understanding of the radiative transfer at Earth's surface. I therefore suspect that the ground-based measurements were not processed correctly, and if so, this would have consequence for the majority of data presented in the manuscript. All affected data would have to be reprocessed. Over the U.S., the OMI overpass occurs approximately between 0 and 2 hours after local solar noon. Hence, the SZA at local solar noon is almost always smaller than at the time of the satellite overpass. UV radiation must therefore be larger, on average, at noon than at the time of the overpass. Of course, UV radiation at the surface during the overpass time may occasionally be larger than during noon, for example, if enhancement by scattered clouds occurs. However it is highly unlikely that atmospheric conditions (e.g. clouds, aerosols, ozone) change in a systematic way between noon and overpass, resulting, on average, in higher absorption at noon to compensate for the smaller SZA. For example, if clouds always had a larger optical depth at noon than at the overpass time, the effect could be explained. It is hardly possible that such a systematic cloud effect could occur for the majority of the 31 sites studied here. Hence, the symmetrical distribution centered at one (Fig. 8b) points to a problem in the data

analysis (perhaps the calculation of the time of local solar noon is incorrect), which must be resolved before the paper can be published.

Response: We thank the reviewer for raising the concern of Fig. 8b in the originally submitted manuscript and we double checked our procedure of processing the ground measured data. It turned out that in our analysis, the local solar noon was defined as the local noon hour that is more or less defined by human and is constant in a time zone of 15-degree-longitude wide over U.S. Since this definition is inconsistent with the OMI algorithm that defines the local solar noon based on the minimax of the local solar zenith angle, we have re-processed the data collocation based on the local solar zenith angle (instead of the local noon hour) to match the noon time estimates by OMI in the revised manuscript. We apologize for our mistakes in the initial analysis. We have recalculated the local solar noon time for each ground site, defined as when the solar zenith angle reaches the minimum during the day. Consequently, the local solar noon time at each of the 31 ground sites in the present manuscript varies with the day of the year. Hence, we have reprocessed all of the data used in the manuscript, reconducted the comparison and updated the manuscript accordingly. The results show that (1) overall, OMI noon-time EDR values are larger than overpass time, (b) the PDF of the ratio is normal and in average 22% of time noon-time values are smaller than OMI overpass time (based on collocated data from ground observations), presumably due to the change of atmospheric conditions, and (c) the correlation of the ratios between OMI and surface data is more reasonble.

My second "major" comment refers to the selection of results being presented. The paper features an abundance of statistical analysis parameters but not the information that most readers would be most interested to see, i.e., the mean bias and range between ground-based measurements and OMI at each site. I therefore propose to include either a new table or a figure with box-whisker plots, which would show for each site the median and average relative bias, the interquartile range, and the 5th and 95th percentile range, for small SZA (e.g., 0 - 50 degree) and large SZA (50 - 75 degree), and for noon and overpass data. Such box-whisker plots would be similar to the box- whiskers shown in Figure 11, and would show data for each of the 31 sites separately, but only for two SZA ranges (which I think is sufficient). Such statistics are much more useful than Taylor diagrams, which are hard to interpret, and separating the results per site would allow to better discuss regional differences in the deviation of ground- based and OMI data, along with their causes (e.g., elevation, proximity to pollution and aerosol sources, clouds, etc.). If a figure with box-whisker plots is chosen, those could be spread-out over two or three rows to be able to show enough detail. Box- whiskers are easier to visually take-in than a table, but a table has the advantage that the numbers are defined. Finally, I don't see much value in Figures 5 and 6 and these could be removed (see details below).

Response: We sincerely thank the reviewer for this comment to improve our manuscript. We have followed the recommendation to generate a panel of box-whisker plots (please see Fig S1. in the supplement). We have also added the relevant discussion in Sect. 4.3. In addition, we have re-arrange and combined some figures to make the results more easy to follow and clear; for example, the PDF analysis is now merged together in Figure 4.

The title of the paper should also mention the ground-based measurements. I propose: "Surface erythemal UV irradiance in the continental United States derived from ground-based and OMI observations: quality assessment, trend analysis, and sampling issues".

Response: We appreciate the reviewer's suggestion for this new title to better represent the current work and we have adopted it.

**Specific Comments**

P1, L17: As mentioned above, I find the conclusion in the following paragraph very surprising and physically impossible

"In addition, the ratio of EDR between solar noon to overpass time is often (95% in frequency) larger than 1 from OMI products; in contrast, this ratio from ground observation is shown to be normally distributed around 1. This contrast suggests that the current OMI surface UV algorithm would not fully represent the real atmosphere with the assumption of a constant atmospheric profile between noon and satellite overpass times."
The summary in the abstract needs to be revised once the issue of the surprising distribution of the ratio of noon and overpass ground-based data is resolved.

Response: We have updated the summary related to Fig 8 in the abstract.

P1, L21: Change "Both OMI Noon_FS and ground peak EDR show a high frequency of occurrence of ~ 20 mW m-2 over the period of 2005–2017. However, another high frequency of ~ 200 mW m-2 occurs in OMI solar noon EDR while the ground peak values show the high frequency around 220 mW m-2, implying that the OMI solar noon time may not always represent the peak daily UV values." to: "The distributions of the OMI Noon_FS and the ground EDR were analyzed using data for the period 2005- 2017. Both distributions have local maxima at about 20 mW m-2. The overall maximum of the distribution is 200 mW m-2 for the OMI and 220 mW m-2 for the ground-based data."

Response: Corrected.

P2, L35: The references of the two assessment reports (UNEP, 2007; WMO, 2010) are now fairly old and should be updated by references to the latest assessment reports (UNEP 2015 and WMO 2014).

Response: Corrected.

P2, L40: Bigelow et al., 1998; Sabburg et al., 2002; Levelt et al., 2006 is an odd collection of rather old papers. Perhaps some newer papers should be cited also.

Response: Thanks for the suggestion and we have added a few newer references.

L55: Explain acronyms UVMRP and USDA.

Response: Corrected.

L56: The Brewer network is also still active in the U.S., see https://www.esrl.noaa.gov/gmd/grad/neubrew/ Please mention it!

Response: We have added this in the text.

L68:     Also mention an example from South American, e.g., http://dx.doi.org/10.1016/j.jphotobiol.2012.06.013

Response: Thanks for the suggestion and we have added this reference.

L142: the standard definition for UV-B is 280-315 nm, not 280-320 nm.

Response: Thanks for the correction.

L146-147: For satellite validation, the uncertainty for SZA > 80° is of lesser importance. What is the uncertainty for the SZA range applicable to OMI validation? (While the specification of deviations from measurements with the standard triad is interesting, the 2.8% specified here is not an uncertainty because measurements of the triad are not free from uncertainty.) Also, please specify the confidence interval of all uncertainty specifications.

Response: We thank the reviewer for asking this question related to the uncertainty of the ground observation data. We rewrote Sect 2.2 (Ground Observation data) with more information about the characterization and calibration of the ground observation data and the associated uncertainty levels.

L148: I don't understand what the authors want to emphasize with the sentence starting with "In spite of this," Do you mean that having small uncertainties is more important when quantifying geographical difference than for satellite validation? If my interpretation is correct, please explain why you think that is the case!

Response: We apologize for the confusion here. We did not mean having small uncertainties is more important when quantifying geographical difference than for satellite validation. We were trying to give an example to show the application of the data. We have removed these few sentences here and instead just focused on discussing the data calibration process and the uncertainty levels.

L175: I am not sure how to interpret "Correspondingly, the temporal mean of ground observation within Delta T is compared to the spatial mean of OMI data within D." As noted earlier, ground based data were aggregated into 3-minute averages. So there can only be 3 ground-based measurements in a +/- 5 min time period. Likewise, There are only one or two OMI measurements within 50 km from the ground site, based on the OMI pixel size discussed earlier. The sentence should be clarified.

Response: We have reorganized the sentence here to make it clear.

L180: Specify whether the number of data pairs refers to all sites or each individual site.

Response: The number of data pairs refers to all sites and we have specified this in the text.

L186-192: Please also provide the formulas for NSD and RMSD. In particular, I am not sure what the difference is between RMSE and RMSD. Equations would clarify this. It is also inconsistent to name the "normalized standard deviation" NSD but the "normalized root-mean-square difference" RMSD. It should be NRMSD. Replace "shown in x and y axis respectively" with "shown both on the x and y axis". Replace "shown as the radius from the expected point" with "shown as concentric circles around the point labeled "expected" in Fig. 3a and 3b.

Response: We apologize for the confusion. The RMSD should be a normalized quantity (please see Eq. (4)). It is the centered root-mean-square difference normalized by the standard deviation of the observational data.

L222: n was already defined as the total number of data points on line 203. Don't use the same symbol for different quantities!

Response: We have corrected this.

L243 and Fig. 2a, b: One would expect that ground-based and OMI measurements agree better at overpass time than noon. This is indeed the case (as described) for MB and RMSE. However, surprisingly, the regression line is closer to the (ideal) 1:1 relationship for Noon_FS data compared to OP_FS data. This observation should be mentioned in the text, and reasons should be given. Perhaps this observation is a result of errors in the processing of ground-based measurements discussed earlier.

Response: We thank the reviewer for the careful observation. The updated figure has shown that the comparison for OP_FS shows better results than that of Noon_FS regarding MB, RMSE, correlation and the slope as well.

Fig. 3 and associated text. I find it very strange that overpass data have a negative bias and noon data a positive one. This points again to a problem in the calculation of one (or both) of the datasets.

Response: Thanks for the good question and we have fixed this issue now. After reprocessing the data, both OMI overpass and local solar noon time comparison show overestimates with OMI overpass time data showing a better agreement with ground observation.

Figure 5 and associated text. While I understand how the figures were calculated, I don't understand why these distributions are important to show. The figures mix in data from all sites (with greatly different cloud and aerosol conditions) and seasons. The figure depends on many variables, which I can't decouple in my head when looking at these distributions, and therefore, I don't know what to learn from these distributions. While there are differences between the distributions for the ground and OMI measurements (which ideally shouldn't exist) I wouldn't be able to grasp from this figure what might have gone wrong with the UV retrieval from OMI data (assuming that the distribution for the ground-based data is correct). I would consider removing this figure and the text that goes with it.

Response: Thanks for the comment and thought. The motive that we show Fig 5 is to illustrate the surface EDR distribution from OMI and ground observations. Previous work by Wang and Christopher (2006) has shown that changes in SZA in a day would cause two peaks in the surface shortwave radiation, one in the morning and one in the late afternoon. Here, we are interested in the surface EDR distribution. The results show that both OMI and ground observations of EDR demonstrate a similar distribution that resembles a bi-lognormal distribution. Therefore, we hope to keep Figure 5 and Figure 6. Further, from a statistical point of view, the comparison of two datasets including bias and PDF is more robust than either alone.

Fig 6 and associated text. Again, I am not sure why this figure is important. I understand that it confirms that OMI surface EDR and ground-observed EDR were drawn from the same distribution, but is this confirmation

really important? What can be learned that is not already known from the correlation coefficient? Hence, I suggest to consider removing this figure also.

Response: Please see the comment above. The linear correlation coefficient is not a statistically robust neither robust nor resistant and certainly it doesn't suggest if the two variables are from the same PDF or not. See Wilks (2011, cited in the manuscript).

L286: define "peak UV", e.g., move or copy "peak refers to the highest dose rate found in a day at each site" from the caption of Fig. 7 to the text.

Response: We have fixed this.

Fig 7 compares two different quantities and it is therefore not surprising that there are differences. For example, the observation of clouds within the OMI pixel will always lead to a reduction of UV radiation because cloud modification factors are <=1. In contrast, ground-based measurements can be enhanced beyond the clear-sky value during broken cloud conditions. This is a well-known phenomenon, e.g., http://doi.org/10.1038/371291a0. It is therefore not surprising that the high-frequency band is at 200 mW m$^{-2}$ for OMI Noon-FS data and 220 mW m$^{-2}$ for "Ground Peak" data. Since it is the goal of the paper to validate OMI data, I see little value in comparing two different quantities and concluding that "OMI solar noon time EDR may not always represent the high peak value on a daily basis due to the varying atmospheric conditions." I suggest replacing the right panel of Fig. 7 with the distribution of Noon_FS data from the ground based measurements.

Response: We thank the reviewer for the explanation of the effects of clouds on both OMI and ground observations. We agree that it helps to compare the same quantity and we have added the comparison with ground Noon_FS (Fig 4). The reason we show the ground peak is to understand if the peak UV always happen at solar noon time. We show that peak UV PDF is different from noontime UV PDF, based solely on the ground-based observation.

L321-325. Results discussed here and in Figure 8 do not make sense to me. Please see my General Comment earlier. For these reasons, I also disagree with the conclusion: "This indicates that the current OMI surface UV algorithm would not fully represent the real atmosphere with the assumption of constant atmospheric conditions being made and could thus induce errors in estimating surface UV irradiances." I suspect a problem with the processing of ground-based data, not of OMI.

Having said this, I note that Bernhard et al. (2015) (see: https://www.atmos-chem- phys.net/15/7391/2015/) have reported a problem in the conversion from OMI overpass EDRs to EDRs at local solar noon by the OMI UV algorithm. They conclude "Additional analysis suggests that the pattern is likely due to a systematic error of up to +/-0.5 degree in the calculation of the local-noon SZA by the [OMI] algorithm. For a SZA of 80 degree, a 0.5 degree error in SZA results in a UVI error of about 8 %". The authors should look at this publication and determined whether this problem also affects their data. According to my judgment, the problem is of less importance for the data of the USDA network than for data from the high-latitude sites discussed by Bernhard et al. (2015) because SZAs at the time of the satellite overpass are much smaller at lower latitudes.

Response: We have replotted Fig 8 with the new data and update the text accordingly. Thanks for sharing the work of Bernhard et al. (2015) that discusses the SZA effects, we agree with the reviewer's judgement that the effects due to the calculation of SZA in the OMI algorithm.

L358: I believe that the larger noise in the bias at larger COT values has additional reasons than the one noted in the paper. For example, it is difficult from space to estimate the COT if the COT is large (e.g., > 10). In contrast, clouds with COT=100 attenuate UV radiation much more than those with COT=10.
Response: Yes, we agree with you. There could be additional reasons than the one discussed in the manuscript. We have revised the statement to make it not being biased.

Figure 13: It would be better to show trends per year or decade in percent instead of absolute values (in mW m$^{-2}$ yr$^{-1}$). The significant trend in one tiny part of California is almost not worth mentioning in the text (L 372).
Response: Thanks for the suggestion. We have shown the trend in percentage per year now instead of absolute values. We also agree that it is not worthy mentioning the tiny part in California.

Figure 14a: Also this plot should be presented in percent. (If you decide to keep trends in absolute terms, the unit should be changed to that of spectral irradiance: mW m$^{-2}$ nm$^{-1}$ yr$^{-1}$.)
Response: Thanks for the suggestion and we have shown the trend in % per year now.

L377: I don't understand why Zhang et al. (2017) found a significant positive trends over the western U.S. using OMI AOD for 2005–2015 while such trend was not found by the same author in the present work. What is the difference? The fact that this study considers measurement between 2005 and 2017 instead of 2005 and 2015?
Response: After recalculating the trends for pixels with at least 20 days of data in a month, we indeed find OMI AOD show some significant trends (Fig S2 (a)) for some pixels in the Western and Central U.S., also found by (Zhang et al., 2017).

L380: Explain acronym "AAOD" I presume it means "absorption aerosol optical depth." Indicate that this parameter is part of the standard OMI data products. AAOD in the UV-B is very difficult to measure from space since most of the absorption is close to the ground. Indicate the uncertainty of OMI AAOD. Could the small AAOD trends (<0.003 per year) that are reported here be a result of drifts in OMI data?
Response: We have described the AAOD in the earlier Sect. 2.1 and we apologize for the many abbreviation and acronyms used in the current manuscript. We have modified Table 2 to better organize the abbreviation we have used in the current manuscript. We have recalculated the AAOD trend (Fig 8(f)) and in most parts, the trend is around 0.05 per year calculated as 100 x AAOD per year. According to the OMI Algorithm Theoretical Basis Document (ATBD) Volume III, the retrieval accuracy of OMI aerosol optical thickness algorithm is on the order of 30 %.

L387: Monthly averages can be greatly biased if only 10 days are available, in particular if measurements occur only at the beginning or end of a month. It would be best to repeat the trend analysis using only months with 20

or more days of data, and compare the results to ensure that trend estimates are robust and not driven by missing days.

Response: We agree that using 10 days could cause biases and we have recalculated the monthly average using only months with at least 20 days of data.

The conclusions need to be changed to reflect changes to the text resulting from my comments above. This applies in particular to lines 437-446 where the distributions of noon/overpass data are discussed.

Response: We have revised this part.

L454-459: The suggestion that trends in UV seen in the ground measurements are caused by trends in AAOD are highly speculative, and this should be stated. I believe that trends in AAOD derived from OMI measurements at 310 nm are rather uncertain. Furthermore, assessments of trends in AOD reported in the paper are only based on OMI (lines 377-379). The conclusions that there are no trends in AOD is premature because of the difficulty to probe the lower troposphere from space in the UV-B. It would be good to check whether the ground-based AERONET sunphotometer network has reported trends in AOD and AAOD for the period of the paper and to use those data to interpret trends in EDRs (although AERONET also does not have wavelengths in the UV-B).

Response: We find significant positive OMI AAOD trends in part of Wester and Central, which could possible affect some of the ground measurements in this region but further analysis would be needed to better understand the cause of the trends found from both ground measurements and OMI data. We agree that it is difficult to probe AOD in the UV-B, especially if the atmosphere is missed with dust and biomass burning aerosols, which make it challenging to study the AOD trend. We strive here to provide some first-order explanation but more detailed analysis would be needed to understand the causes of the surface UV trends. We found that there is no scientifically sound and coherent trends among OMI data for aerosols, clouds, and ozone that can explain the surface UV trends revealed either by OMI or ground-based estimates; nor these data can reconcile trend differences between the two estimates.

**Technical Corrections**

General: It would be helpful to define acronyms for "overpass time EDR" and "local solar noon EDR" at the beginning and then use these acronyms in the text instead of repeating the same phrases.

P1, L13: Change "OMI data overall has ~4% underestimate for overpass EDR while ~8% overestimate for the solar noon time EDR" to "OMI underestimates the overpass EDR by about 4% on average and overestimates the solar noon time EDR by 8%."

P1, L17: "with SZA" > "for SZA"

Response: Corrected.

P1, L20: "viability" > "variability"

Response: Corrected.

P1, L21: Explain "Noon_FS"

L44: "in many locations" > "at many locations

Response: Corrected.

L47: "In addition, the surface UV irradiance, denoted as 'erythmal weighted'" > "In addition, the erythemally weighted irradiance" (spectral irradiance is also surface UV irradiance in this context!)

Response: Corrected.

L49: "the incoming solar radiation" > "the solar irradiance"; and "according to" > "with"

Response: Corrected.

L51: "derive UV index" > "derive the UV index"

Response: Corrected.

L52: "UV index" > "the UV index"

Response: Corrected.

L60: "has been" > "have been" (data is plural) L61: "the OMI data has" > "OMI data have"

Response: Corrected.

L64: "in many sites" > "at many sites"

Response: Corrected.

L100: "tables are the" > "tables is the"

Response: Corrected.

L119: "triangular slit function with full width half maximum" > "a triangular slit function with full width at half maximum"

Response: Corrected.

L177: delete "different"

Response: Corrected.

L187: "normalized room-mean-square" > "normalized root-mean-square" (not "room"!) L262 "larger" > "large"

Response: Corrected.

L364: "range principally controlled" > "range that is greatly affected"

Response: Corrected.

L366: "identified trend of surface EDR" > "trends in surface EDR"

Response: Corrected.

L388: "In contrast, ground observation shows" > "In contrast to trends derived from OMI data, ground observations show"

Response: Corrected.

L425: Explain acronyms "TEMPO and GEMS"

Response: Corrected.

---

## Author Comment (AC3) · 23 Dec 2018

We sincerely thank the editor and reviewers for taking the time to review our manuscript and providing constructive feedback to improve our manuscript. We have revised the manuscript accordingly by following the reviewers' suggestion. Below shown are the original comments from reviewers in black and our corresponding responses in blue.

**Comments by RC3:**

The manuscript by Zhang et al., "OMI surface UV irradiance in the continental United States: quality assessment, trend analysis, and sampling issues", presents a study to evaluate the estimates of surface UV from OMI satellite measurements against ground- based measurements and additionally carries out a trend analysis. There have been several validation studies of OMI surface UV before, unfortunately it remained unclear what is the understanding that this manuscript adds to the current knowledge.

Nothing was said about the quality and uncertainties of the ground-based measurements. And since the results were peculiar enough, it was not clear whether the ground measurements tell something about OMI UV performance or whether it is other way around: comparison against OMI UV tells something about the systematic errors in the ground-based measurements. In my opinion, it takes a major revision before the manuscript can be fully evaluated. I will explain and clarify these points below.

Response: We thank you for your time reviewing the paper. As noted in the introduction (and other reviewers), prior to this study there were few studies that conducted systematic evaluation of OMI surface erythemal UV data with long-term ground-based observations in the U.S. There is also unavoidable assumption that is made in the OMI UV algorithm to derive surface UV at local noon time from satellite overpass time. This assumption needs to be evaluated and the associated uncertainties should be assessed as the local noon time is expected to have the largest surface UV radiation relevant for health studies. Furthermore, in our revised manuscript, we also emphasize the analysis of surface observation to illustrate that the peak UV at the surface is not always at local solar noon. We also assessed the OMI surface UV data in terms of their PDFs (that going beyond linear coefficient and bias). While we agree that both surface measurements and satellite measurements have uncertainties, in the revision we have emphasized the discrepancy at noon is larger than the uncertainty in surface measurements, highlighting the likelihood of sampling issues related to the inherent limitations of polar-orbing satellite such as OMI. The comparison helps us define the uncertainty enveloped in long-term estimates of the surface UV radiation from both in situ and satellite observations, assess if there are any coherent trends and differences between the two datasets, and recommend possible improvements for OMI and the need of geostationary measurements of surface UV. The last but not learn, we also show that there is no scientifically sound and coherent trends among OMI data for aerosols, clouds, and ozone that can explain the surface UV trends revealed either by OMI or ground-based estimates; nor these data can reconcile trend differences between the two estimates.

**Main comments:**

As the authors quite correctly mention, the information about the atmospheric conditions comes from the OMI measurement at the overpass time. So this statement is already suggesting that only the overpass time comparisons are meaningful. And indeed, my suggestion is to exclude the local noon time comparisons. However, since the authors presented also the local noon time comparisons with strange results, I want to discuss these results briefly below.

Response: We thank the reviewer for the suggestion and the comments. We understand that the comparison for overpass time would be more accurate. However, the surface UV irradiance at local solar noon time is also an important quantity as it is expected to be the largest during a day and is relevant to health studies. There is a need to quantify how much errors would be, because of the assumption of constant atmospheric conditions between satellite overpass time and the local solar noon time. Hence, we hope to keep the comparison for both satellite overpass and local solar noon time, in order to shed some light on the uncertainties from inherent limitations in the sampling made by polar-orbiting satellite. Our evaluation here not only reveals how good the OMI product is - its PDF for surface UV is in statistically significant agreement with the counterpart of surface observation, but also quantifies the uncertainties in creating surface peak UV climatology from a polar-orbiting satellite (such as OMI) that has inherent limitations in sampling. The latter is analysed here by taking advantage of long-term calibrated 3-minute surface UV data record over the U.S.

The SZA is always somewhat lower at the local noon time than at the overpass time, resulting in larger UV irradiance at the local noon by this SZA effect. This means that there has to be some very clear and systematic effect in cloudiness to compensate this, if the results in the Figure 8b are true. Cloud effect would the only plausible explanation and it would mean that at the OMI overpass time there should be systematically (on average) lower cloud amount than at the local solar noon over the stations studied. From the results of Meskhidze et al. 2009, the opposite diurnal cloud effect seems to be the case, on average thicker (higher cloud optical depth) clouds in the afternoon than before (at Terra overpass time) over most of the continental US. The overpass time of Terra is before noon, but I think this should give an indication, at least, about the sign of the systematic difference between noon and OMI overpass. So how do you explain your Figure 8b? It is absolutely necessary to include ancillary data or publication citation to convince the reader about the behavior in the Figure 8b. Otherwise, he/she is left with an impression that something has to be wrong with the ground-based measurements.

Response: We fully agree with the reviewer's point, and that is indeed one of our motivations to look into the issue: to what extent a constant atmospheric profile between the local solar noon and satellite overpass time is valid. Following your suggestion as well as the suggestion made by the second reviewer, we have double checked our analysis and found we mis-calculated the local solar noon time. Subsequently, we make corrections in the revisions. The results are interesting. Overall, OMI local solar noon time are about 95% times larger than satellite overpass time; but still, in average 22% of times, the ground measurements show that the local solar noon time surface UV is smaller than that in satellite overpass time, likely reflecting the differences in variations of clouds and other parameters (Meskhidze et al., 2009).

If the results in the Figure 8b are not understood and explained, the reader can only assume that there has to be some angular dependent error left in the measurements to cause this. And indeed, there was no single word about the uncertainty of the ground-based measurements. The reference to Lantz et al. 1999 was just given, without any further discussion. So please explain in detail the corrections applied. The correction should be ozone and SZA dependent, as they discuss in Lantz et al. 1999. From where you take the ozone values for the correction? How often the reference is calibrated? How often these instruments participate in inter-comparison campaigns? I think it would be an informative plot to show also the typical correction factor, plotted as a function of SZA, for two very different ozone

amounts. The angular calibration is also a function of cloudiness, so please discuss in detail how it is included in the correction of ground-based measurements.

Response: We thank the reviewer for this specific comment related to the quality of the ground observational data used in the current work. We have revised Sect. 2.2 (ground observation data) to provide more information about the calibration and characterization process of the UMMRP broadband radiometers used in this work. The UMMRP uncertainty is ozone and SZA dependent. The erythemal UV irradiance used in the current work is prepared with the calibration factors of SZA dependence that assume the total column ozone is 300 DU. This is the best data offered by UMMRP to the scientific community in the public domain. We understand that the variation of $O_3$ amount may lead to errors in the calibration, and in the long-term, there are conditions where $O_3$ amount can be larger than 300 DU, and there are also times where $O_3$ amount is less than 300 DU. Kimlin et al. (2005) has shown overall UMMRP broadband radiometers have an uncertainty on the average of $\pm$ 6%.

Currently the trend analysis part does not offer anything consistent. For the careful reader, the main message seems to be that the ground-based measurements result in both negative and positive trends, no matter how the data are selected, and even for stations that are almost side by side. So perhaps this trend analysis could be also excluded, or the authors explain what is the consistent message it brings. For instance, let's take a look at the East coast of US, some sites give slightly negative trend (13b), the southern most shows a positive trend. If one studies Zhang et al. 2017 in detail, it is clear that there is no AAOD trend in this region, while there is a negative trend in AOD (from OMI, but also from other instruments). So one would assume slight positive trend in surface UV, but in the contrary there is negative trend in two sites.

Response: The trends derived from ground measurements are within the measurement uncertainty range. Our original hypothesis is to see negative trend because of recovery of $O_3$, but on the other hand, aerosols amounts are also declining. Our analysis eventually proves the null hypothesis there is no coherent trend. Indeed, we found there is no scientifically sound and coherent trends among OMI data for aerosols, clouds, and ozone that can explain the surface UV trends revealed either by OMI or ground-based estimates; nor these data can reconcile trend differences between the two estimates. While it would be nice and interesting to see a coherent trend, proving a null-hypothesis is equally scientifically important.

The changes in AAOD alone cannot explain these trends, so it is absolutely important to consider the simultaneous changes in AOD as well. If one selects those regions from Zhang et al. 2017 where both AOD and AAOD (from OMI) shows positive change, then the most probable sites are Holtville, CA and Las Cruces, NM, where based on this change, one would assume to see negative change in surface UV. However, both stations show positive change. These are just two examples why I argue that in its current form, the manuscript fails to offer consistent and convincing message about the trend analysis. My two main comments might be linked: if the quality of the ground-based measurements is not sufficient and/or properly considered, then these issues might become visible both in the results shown in Figure 8b and also in the trend analysis. It can be also, that in any case, the signal is too weak to detect any meaningful trend, but if so, then the discussion and comparison against OMI AAOD is not justified.

Response: We agree that the key message should be clearly delivered. Overall findings are as followings:

1) at satellite overpass time, 7 % overestimation by OMI in comparison with surface measurement, and this bias, while close to, is definitely larger than 6 % uncertainty range of surface measurements.

2) at local noon time, in average, OMI noon time EDR is 18 % larger than at satellite overpass time, but 22 % of times, noontime is smaller derived from the ground measurements.

3) at local noon time, the satellite-based OMI is 7% bias higher than surface counterpart, and this bias is out of the surface measurement range.

4) we find several sides shows statistically significant trend, but the trend magnitude overall is within the surface measurement range. There is no scientifically sound and coherent trends among OMI data for aerosols, clouds, and ozone that can explain the surface UV trends revealed either by OMI or ground-based estimates; nor these data can reconcile trend differences between the two estimates. While it would be nice and interesting to see a coherent trend, proving a null-hypothesis is equally scientifically important

**Specific comments:**

Line 364, you mention that the absorbing aerosols could be the reason for the OMI UV trend. It is not the likely reason, since the correction is taken from monthly climatology. So what did you mean?

Response: We agree with the reviewer that the absorbing aerosols will not be likely the reason for the OMI UV trend detected. Other factors such as the scattering aerosols concentrations, clouds and trace gas concentrations could all contribute to the trend. The key message here is that on the first order, there is no apparent trend in the factors that affect the surface UV, and there is also no apparent and spatially coherent trend in OMI UV data.

Line 370, there is a better reference to Kinne et al. (Kinne et al. 2013 below).

Response: Thanks for the suggestion and we have updated the reference.

Line 361, if you include 310nm, perhaps comparison to the 380nm trend would bring something useful, since it does not have any significant ozone absorption.

Response: Thanks for the good suggestion and we have also studied the trend of OMI spectral irradiance at 380nm (please see Fig 8(e)).

Table 3. If you used ordinary least squares for regression, please remember that it gives a systematically biased slope when there is uncertainty in x-axis. These numbers, slopes in particular, would be informative if the method for the regression is correct one. See Cantrell et al. 2009 or Pitkanen et al. 2016. Please explain the method that was used and the possible limitations.

Response: Thanks for bring us the attention of those two references. We have explained the regression method used in the current work and future work should be paying more attention to the linear regression methods used as discussed in these two reference papers.

REFERENCES:

Cantrell, C. A.: Technical Note: Review of methods for linear least-squares fitting of data and application to atmospheric chemistry problems, Atmos. Chem. Phys., 8, 5477-5487, https://doi.org/10.5194/acp-8-5477-2008, 2008.

Kinne, S., D. O'Donnel, P. Stier, S. Kloster, K. Zhang, H. Schmidt, S. Rast, M. Gior- getta, T. F. Eck, and B. Stevens, MAC-v1: A new global aerosol climatology for climate studies, J. Adv. Model. Earth Syst., 5, 704–740, doi: 10.1002/jame.20035, 2013.

Meskhidze, N., Remer, L. A., Platnick, S., Negrón Juárez, R., Lichtenberger, A. M., and Aiyyer, A. R.:Exploring the differences in cloud properties observed by the Terra and Aqua MODIS Sensors, Atmos. Chem. Phys., 9, 3461-3475, https://doi.org/10.5194/acp-9-3461-2009,2009.

Pitkänen, M. R. A., S. Mikkonen, K. E. J. Lehtinen, A. Lipponen, and A. Arola, Artificial bias typically neglected in comparisons of uncertain atmospheric data, Geophys. Res. Lett., 43, 10,003–10,011, doi: 10.1002/2016GL070852, 2016.

**References**

Bernhard, G., Arola, A., Dahlback, A., Fioletov, V., Heikkilä, A., Johnsen, B., Koskela, T., Lakkala, K., Svendby, T., and Tamminen, J.: Comparison of OMI UV observations with ground-based measurements at high northern latitudes, Atmospheric Chemistry and Physics, 15, 7391-7412, 2015.

Kimlin, M. G., Slusser, J. R., Schallhorn, K. A., Lantz, K. O., and Meltzer, R. S.: Comparison of ultraviolet data from colocated instruments from the US EPA Brewer Spectrophotometer Network and the US Department of Agriculture UV-B Monitoring and Research Program, Optical Engineering, 44, 041009, 2005.

Meskhidze, N., Remer, L., Platnick, S., Negrón Juárez, R., Lichtenberger, A., and Aiyyer, A.: Exploring the differences in cloud properties observed by the Terra and Aqua MODIS Sensors, Atmos. Chem. Phys, 9, 3461-3475, 2009.

Pitkänen, M. R., Mikkonen, S., Lehtinen, K. E., Lipponen, A., and Arola, A.: Artificial bias typically neglected in comparisons of uncertain atmospheric data, Geophysical Research Letters, 43, 10,003-010,011, 2016.

Wang, J., and Christopher, S. A.: Mesoscale modeling of Central American smoke transport to the United States: 2. Smoke radiative impact on regional surface energy budget and boundary layer evolution, Journal of Geophysical Research: Atmospheres, 111, 2006.

Zhang, L., Henze, D. K., Grell, G. A., Torres, O., Jethva, H., and Lamsal, L. N.: What factors control the trend of increasing AAOD over the United States in the last decade?, Journal of Geophysical Research: Atmospheres, 122, 1797-1810, 2017.

---

## Author Response (AR2)

**Response to the editors and reviewers:**

Comments:

What I am saying is that the whole paper should be more focused (abstract, figures, conclusions) in the overpass data. It would be useful to highlight that there is a X% overall overestimation of OMI compared with ground for overpass spectra and a table or figure to show this overestimation ( for example as a median of the ratio OMI vs ground ) plus the standard deviations etc. So this is the main finding. Then for the noon measurements someone can say that what we expect is more scatter due to the OMI assumptions that have been discussed above.

So summarizing, to focus more on the overpass results and description and less on the local noon ones (that can be also presented). Temporal and spatial analysis and trends are also important as presented .

Replies

The statistics for evaluating OMI data at its overpassing time is presented in the abstract and several figures in the manuscript. While this is important, it is only one aspect of the finding of this paper. Another aspect of this paper, which in our view is more interesting, is the study of noontime surface UV data based on surface observation data, and how the OMI noontime estimate can have errors due to temporal sampling bias. OMI has been around for more than a decade, and noontime UV estimate is routinely generated as a scientific parameter for the use in the community. Therefore, it is important to assess how good it is and the extent to which the temporal sampling issue can lead to biases. To take the reviewer's point, we now have also added in the appendix on the statistics at each station for the comparison between OMI and surface data at satellite overpassing time. There are good sciences that OMI has to miss due to its limited once-per-day sampling. The paper is now in good balance of using surface observation data and satellite UV data; the title is also changed to add surface observations as suggested in the previous review. Thanks.

Minor

Line 172

there is also this new reference to have a look

Zempila, M.M., et al ., Validation of OMI erythemal doses with multi-sensor ground-based measurements in Thessaloniki, Greece, Atmospheric Environment, doi: 10.1016/j.atmosenv.2018.04.012, 2018

and line 172.

This one for the spatial and temporal issues:

S. Kazadzis, A. Bais, D. Balis, N. Kouremeti, M. Zempila, A. Arola, E. Giannakaki, A. Kazantzidis, V. Amiridis, Spatial and

temporal UV irradiance and aerosol variability within the area of an OMI satellite pixel, Atm. Chem. and Phys., 9, 7273-7298, 2009

Replies. Done. These two references are added and discussed in places around L172.